# FLEXLoRA: ENTROPY-GUIDED FLEXIBLE LOW-RANK ADAPTATION

**Muqing Liu**[1,†], **Chongjie Si**[2,†], **Yuheng Jia**[3,4*]
[1]Chien-Shiung Wu College, Southeast University [2]MoE Key Lab of Artificial Intelligence,
AI Institute, School of Computer Science, Shanghai Jiao Tong University [3]School of
Computer Science and Engineering, Southeast University [4]Key Laboratory of New Generation
Artificial Intelligence Technology and Its Interdisciplinary Applications (Southeast University),
Ministry of Education, China
liumq23@seu.edu.cn      chongjiesi@sjtu.edu.cn      yhjia@seu.edu.cn

## ABSTRACT

Large pre-trained models achieve remarkable success across diverse domains, yet fully fine-tuning incurs prohibitive computational and memory costs. Parameter-efficient fine-tuning (PEFT) has thus become a mainstream paradigm. Among them, Low-Rank Adaptation (LoRA) introduces trainable low-rank matrices and shows strong performance, nevertheless, its fixed-rank design limits flexibility. Dynamic rank allocation methods mitigate this issue by pruning redundant directions; however, they often rely on heuristic, element-level metrics that globally sort rank directions without matrix-wise distinction, and they lack mechanisms to expand capacity in layers requiring additional adaptation. To overcome these limitations, we propose FlexLoRA, an entropy-guided flexible low-rank adaptation framework that (i) evaluates matrix importance via spectral energy entropy, (ii) supports rank pruning and expansion under a global budget, and (iii) employs zero-impact initialization for newly added singular directions to ensure stability. By addressing granularity, flexibility, and stability limitations, FlexLoRA provides a more principled solution for PEFT. Extensive experiments show that FlexLoRA consistently outperforms state-of-the-art baselines across benchmarks.

## 1 INTRODUCTION

Since large pre-trained models have advanced the state of the art in numerous tasks (Kirillov et al., 2023; Devlin et al., 2018; Liu et al., 2019; Peng et al., 2025), adapting them to downstream tasks has become a prevailing way recently. However, their adaptation requires fully fine-tuning, i.e., updating all parameters, which incurs substantial computational and memory costs (Ma et al., 2024; Raffel et al., 2020; Qiu et al., 2020). To address this challenge, parameter-efficient fine-tuning (PEFT) methods have been proposed (Zhang et al., 2022a; Si et al., 2024; Pfeiffer et al., 2020; Houlsby et al., 2019; Hu et al., 2021; He et al., 2021a), which adapt pre-trained models by updating only a small subset of parameters while preserving competitive performance.

Among PEFT approaches, Low-Rank Adaptation (LoRA) (Hu et al., 2021) and its variants (Zhang et al., 2022a; Wu et al., 2024) have emerged as representatives, which introduces trainable low-rank matrices to approximate task-specific updates of pre-trained weights. Despite its effectiveness, LoRA employs fixed-rank across all layers, thereby limiting flexibility in allocating model capacity. To address this issue, a series of dynamic rank allocation methods have been proposed, such as AdaLoRA (Zhang et al., 2023), SaLoRA (Hu et al., 2023a), and AutoLoRA (Zhang et al., 2024). These methods usually compute a heuristic importance score such as parameter gradient for each individual rank direction. The scores from all ranks across all matrices are then aggregated and globally sorted, after which the least important directions are pruned. These strategies partially alleviate the limitations of LoRA.

---

*Corresponding author.
†Equal contribution.

However, these dynamic rank allocation methods still suffers from three key limitations. First, the importance metrics are typically heuristic, relying on approximations such as parameter sensitivity rather than principled criteria. Second, all rank directions from different matrices are globally sorted and pruned together, ignoring matrix-level distinctions and thereby risking the removal of structurally important directions. Third, the allocation is unidirectional, as it only prunes redundant ranks without mechanisms to expand capacity in layers that demand additional expressive power. Together, these limitations hinder the ability of existing methods to allocate model capacity in a principled and adaptive manner, motivating the need for a more flexible framework.

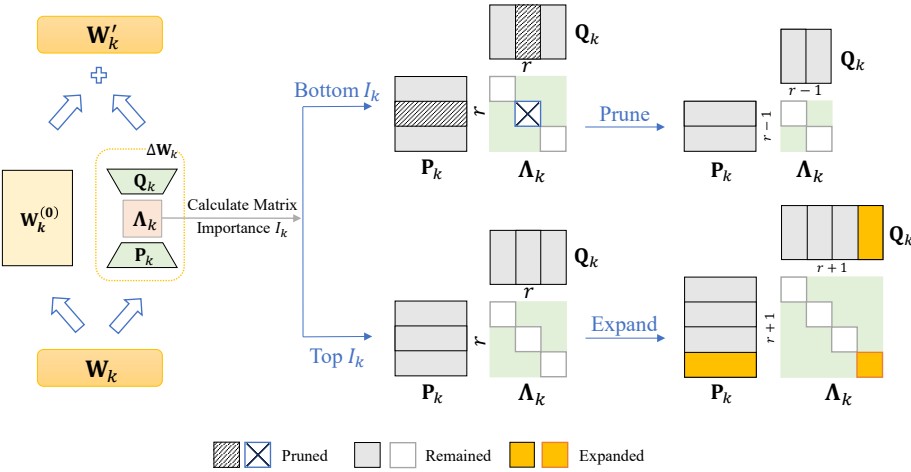

Figure 1: Framework of FlexLoRA. For each weight matrix $\mathbf{W}_k$, FlexLoRA represents the update in an SVD-like form $\Delta\mathbf{W} = \mathbf{P}_k\mathbf{\Lambda}_k\mathbf{Q}_k$, where $\mathbf{\Lambda}_k$ is a diagonal matrix. It then computes a spectral entropy–based importance score for each $\Delta\mathbf{W}$. All scores are globally ranked under a given rank budget: matrices with lower scores prune the least significant direction in $\mathbf{\Lambda}_k$, while those with higher scores receive additional ranks. The newly allocated ranks are initialized with a zero-impact scheme to preserve the original input while enabling subsequent learning.

To this end, we propose FlexLoRA, a novel dynamic low-rank adaptation framework. FlexLoRA reallocates computational resources to the most critical layers, ensuring sufficient capacity for important layers while removing redundancy from less important ones. Specifically, it evaluates the importance of each low-rank matrix at the matrix level using spectral entropy and dynamically adjusts the rank allocation accordingly: pruning the least significant directions in low-importance layers while expanding the rank in layers that demand additional capacity. For expansion, FlexLoRA adopts a zero-impact initialization strategy, where newly added singular directions are initialized with zero singular values and Gaussian-sampled singular vectors, thereby preserving the original input while enabling stable training. Extensive experiments across diverse benchmarks demonstrate that FlexLoRA consistently outperforms strong PEFT baselines under the same parameter budget.

Our contributions are as follows:

- We propose FlexLoRA, a novel framework that supports both pruning and expansion of ranks under a global budget, enabling dynamic reallocation of model capacity across layers.

- We introduces a spectral entropy–based criterion to assess the importance of low-rank matrices at the matrix level, overcoming the limitations of heuristic, element-wise metrics.

- Extensive experiments demonstrate that FlexLoRA achieves superior performance compared with state-of-the-art PEFT baselines under identical parameter budgets.

## 2 RELATED WORK

### 2.1 PARAMETER-EFFICIENT FINE-TUNING

Large pre-trained models exhibit strong generalization across diverse downstream tasks (Xu et al., 2023; Han et al., 2024; Lin et al., 2024; Wu et al., 2025), yet full fine-tuning remains computationally costly and storage-inefficient. Parameter-efficient fine-tuning (PEFT) methods address this issue by reducing trainable parameters while retaining most of the performance of full fine-tuning. Early PEFT approaches fall into three main categories: (1) **adapter-based methods** (Houlsby et al., 2019; Chen et al., 2022; Luo et al., 2023; Pfeiffer et al., 2020; He et al., 2021a; Mahabadi et al., 2021), which insert lightweight trainable modules; (2) **prompt-based methods** (Lester et al., 2021; Razdaibiedina et al., 2023; Wang et al., 2023; Fischer et al., 2024; Yang et al., 2023), which learn task-specific prompt vectors; and (3) **low-rank methods** (Hu et al., 2021; Liu et al., 2024; Zhang et al., 2022a; Kopiczko et al., 2023; Qiu et al., 2023; Renduchintala et al., 2023; Wang et al., 2024), which approximate parameter updates via low-rank factorization.

Among these, LoRA (Hu et al., 2021) is the most widely adopted due to its simplicity and effectiveness. By injecting trainable low-rank matrices into linear transformations, LoRA achieves competitive performance with far fewer trainable parameters. However, its fixed-rank design across all layers limits flexibility and prevents adaptive capacity allocation to task-specific requirements.

### 2.2 LoRA VARIANTS WITH DYNAMIC RANK ALLOCATION

To overcome the limitation of fixed-rank LoRA, recent studies have explored dynamic rank allocation, where the rank of each LoRA module is adaptively adjusted during training. Existing approaches can be grouped into three families. (1) SVD-driven allocation: AdaLoRA (Zhang et al., 2023) and SaLoRA (Hu et al., 2023a) periodically decompose low-rank matrices and prune less important singular directions while reallocating capacity to critical layers. (2) Singular rank decomposition (SRD): DoRA (Mao et al., 2024), AutoLoRA (Zhang et al., 2024), and SoRA (Ding et al., 2023) dynamically split or merge singular components based on gradient statistics or structural priors. (3) Sampling-based allocation: DyLoRA (Valipour et al., 2022) and QDyLoRA (Rajabzadeh et al., 2024) stochastically vary ranks across iterations to improve robustness through randomization. Besides rank-adjusting approaches, methods like MLAE (Wang et al., 2024) decompose low-rank matrices into rank-1 experts and apply expert-level stochastic masking, implicitly modulating effective capacity without altering the nominal rank, thus offering a complementary perspective to dynamic-rank LoRA variants. The effectiveness of these methods largely depends on heuristic sensitivity-based metrics, which aggregate gradient–weight products across singular directions (Liang et al., 2021; Sanh et al., 2020; Zhang et al., 2022b) and smoothed with moving averages or uncertainty terms.

Despite their progress, these strategies remain component-level heuristics: they estimate the importance of each singular direction or parameter independently, while neglecting the structural interactions at the matrix level. As a result, they are prone to gradient noise, lack stability, and may overlook the coordinated role of singular directions within the entire matrix, leading to suboptimal rank adjustments.

### 2.3 ENTROPY-GUIDED METRICS

Entropy, a fundamental concept in information theory for quantifying uncertainty (Shannon, 1948; Jaynes, 1957), has recently been adopted to measure redundancy and information content in neural networks (Achille & Soatto, 2017). Entropy-based criteria have also been applied to pruning and compression, guiding the removal of uninformative components while preserving capacity (Liao et al., 2024). These works highlight that entropy is not only a statistical measure of uncertainty but also a principled indicator of representational richness, providing insight into the information distribution of entire matrices rather than individual parameters.

## 3 METHOD

We propose FlexLoRA, a flexible low-rank adaptation framework that dynamically allocates ranks across layers of large pre-trained models. FlexLoRA consists of three key components: (i) a matrix-level entropy-guided importance metric, (ii) rank pruning and expansion under a global budget, and (iii) zero-impact initialization. We first introduce the basic formulation of FlexLoRA, and then present our method in detail in the subsequent subsections.

### 3.1 SVD-BASED LOW-RANK ADAPTATION

Low-Rank Adaptation (LoRA) (Hu et al., 2021) represents task-specific updates to a pre-trained weight matrix $\mathbf{W} \in \mathbb{R}^{d_{\text{out}} \times d_{\text{in}}}$ using two trainable low-rank matrices:

$$\Delta \mathbf{W} = \mathbf{BA}, \quad \mathbf{A} \in \mathbb{R}^{r \times d_{\text{in}}}, \ \mathbf{B} \in \mathbb{R}^{d_{\text{out}} \times r}, \tag{1}$$

where $r \ll \min(d_{\text{out}}, d_{\text{in}})$ is a small fixed rank to ensure parameter efficiency. Here, $\Delta \mathbf{W}$ serves as a low-rank update that is added to the frozen pre-trained weights, yielding the effective parameter matrix $\mathbf{W}' = \mathbf{W} + \Delta \mathbf{W}$. Similar to LoRA's formulation and those in prior works (Zhang et al., 2023; Meng et al., 2024), FlexLoRA adopts an SVD-based formulation, where $\Delta \mathbf{W} = \mathbf{P \Lambda Q}$. Here, $\mathbf{P} \in \mathbb{R}^{d_{\text{out}} \times r}$ and $\mathbf{Q} \in \mathbb{R}^{r \times d_{\text{in}}}$ represent singular vectors, and $\mathbf{\Lambda} = \text{diag}(\lambda_1, \lambda_2, \dots, \lambda_r) \in \mathbb{R}^{r \times r}$ is a diagonal matrix of singular values. To maintain stability and preserve the SVD property ($\mathbf{P}^\top \mathbf{P} = \mathbf{I}$, $\mathbf{QQ}^\top = \mathbf{I}$), we introduce an orthogonality regularization term:

$$\mathcal{R}(\mathbf{P}, \mathbf{Q}) = \|\mathbf{P}^\top \mathbf{P} - \mathbf{I}\|_F^2 + \|\mathbf{QQ}^\top - \mathbf{I}\|_F^2, \tag{2}$$

where $\| \cdot \|_F$ is the Frobenius norm. However, the rank $r$ of LoRA and its variants is fixed across layers, which may hinder adaptation flexibility and prevent efficient utilization of model capacity. To overcome this limitation, FlexLoRA introduces three synergistic mechanisms that enable dynamic adjustment of the effective rank and maximize its utilization.

### 3.2 MATRIX-LEVEL ENTROPY-GUIDED IMPORTANCE METRIC

A key challenge in dynamic rank allocation lies in determining the importance of singular directions. As discussed in Related Work (Sec. 2.2), prior methods largely rely on sensitivity-based heuristics that design importance metrics at the level of individual parameters or singular directions. Such approaches overlook the structure of the entire matrix, leading to noisy optimization.

To overcome these limitations, we propose a *matrix-level* entropy-guided importance metric. Unlike sensitivity-based measures, entropy-based evaluation captures the intrinsic geometry of the matrix throughout training. Given singular values $\mathbf{\Lambda}$, the spectral entropy importance score is defined as

$$I(\mathbf{\Lambda}) = -\frac{1}{\log r} \sum_{i=1}^{r} s_i \log(s_i + \epsilon), \quad s_i = \frac{\lambda_i^2}{\sum_{j=1}^{r} \lambda_j^2}, \tag{3}$$

where $\epsilon$ is a small constant to avoid numerical issues. The entropy is normalized by $\log r$ to ensure that the spectral entropy is bounded within $[0, 1]$ and comparable across different ranks; the rationale for this normalization is provided in Appendix B. Intuitively, a low entropy indicates that energy is concentrated in a few singular values, suggesting redundancy and suitability for pruning, while high entropy reflects a more balanced distribution, implying richer structural capacity.

### 3.3 RANK PRUNE AND EXPANSION WITH ZERO-IMPACT INITIALIZATION

With the spectral-entropy–based confidence score in place, we can dynamically adjust the rank of each matrix during training. Unlike prior approaches that primarily focus on rank reduction, our strategy supports both expansion and pruning, enabling more flexible capacity reallocation. Specifically, at each training step $t$, we define a rank budget $b(t)$, which specifies the maximum number of singular directions that can be either added or removed in that step (the detailed design of $b(t)$ is described in Sec. 4.2). This budget acts as a global constraint, ensuring that rank adjustments remain stable and computationally tractable while still allowing sufficient adaptivity.

To remove redundancy, we first rank matrices by their importance score and identify the $b(t)$ least important ones with rank greater than one. Within each selected matrix, we reduce the rank by discarding the singular component associated with the smallest singular value. This choice is grounded in the SVD principle that directions with minimal singular values contribute least to the matrix's representational power. Moreover, the corresponding importance score $I(\lambda_{\min})$ is also the smallest, indicating minimal spectral contribution; a formal proof of this monotonicity is provided in Appendix C. For expansion, we select the $b(t)$ most important matrices. Within each selected matrix, we add a new singular direction with its singular value initialized to zero, while the corresponding vectors are sampled from Gaussian distribution. This zero-impact initialization ensures that the insertion does not perturb the current output and allows the new direction to be gradually optimized during training, which guarantees stability while enabling effective capacity growth when needed.

By leveraging the matrix-level spectral entropy as a confidence score, FlexLoRA adaptively adjusts ranks through both pruning and expansion, while zero-impact initialization ensures stable incorporation of new capacity. This joint design enables FlexLoRA to allocate model capacity more flexibly across layers, preserving redundancy-free directions while amplifying structurally informative ones. The full procedure is summarized in Algorithm 1.

---

**Algorithm 1** FlexLoRA

---

**Require:** $S$: total steps, $T$: bidirectional rank allocation period, $r$: initial rank,
**Require:** $b$: total ranks pruned/expanded per step
1: **Initialize:** $(\mathbf{P}_{0:N-1}, \mathbf{\Lambda}_{0:N-1}, \mathbf{Q}_{0:N-1})$ for all weight matrices with rank $r$
2: **for** $i = 1$ to $S$ **do**
3:      UPDATEWEIGHTS($\mathbf{P}, \mathbf{\Lambda}, \mathbf{Q}$)
4:      $I_{0:N-1} \leftarrow$ CALCULATEIMPORTANCE($\mathbf{\Lambda}$)
5:      **if** $i \in T$ **then**
6:          $L \leftarrow$ LEASTIMPORTANTMATRICES($I, b$)      *Identify top-b **least** important matrices*
7:          $M \leftarrow$ MOSTIMPORTANTMATRICES($I, b$)      *Identify top-b **most** important matrices*
8:          PRUNERANKS($L, \mathbf{P}, \mathbf{\Lambda}, \mathbf{Q}$)
9:          EXPANDRANKS($M, \mathbf{P}, \mathbf{\Lambda}, \mathbf{Q}$)
10:      **end if**
11: **end for**
12: **return** Fine-tuned parameters $(\mathbf{P}, \mathbf{\Lambda}, \mathbf{Q})$

---

# 4 EXPERIMENT

## 4.1 MODELS, DATASETS AND BASELINES

We evaluate FlexLoRA on both language and vision models on three tasks to demonstrate its generality. For natural language processing, we adopt DeBERTaV3-base (He et al., 2021b), a widely used encoder model for understanding tasks, and the LLaMA family of large language models (AI@Meta, 2024), which represent the current state-of-the-art in generative modeling. For computer vision, we extend FlexLoRA to transformer-based backbones and assess its adaptability to recognition tasks.

We consider three representative categories of benchmarks:

- Natural Language Understanding (NLU): We use the GLUE benchmark (Wang et al., 2018), which includes diverse sentence- and pair-level classification tasks.

- Commonsense Reasoning (CR): We evaluate on eight widely used benchmarks covering diverse reasoning forms, including yes/no question answering (BoolQ), physical and social reasoning (PIQA, SIQA), narrative completion (HellaSwag), pronoun resolution (WinoGrande), and multiple-choice knowledge-intensive reasoning (ARC-e, ARC-c, OBQA), as well as additional tasks such as CommonsenseQA and SocialIQA (Sap et al., 2020).

- Visual Recognition (Vision):To test generality beyond NLP, we employ the Visual Task Adaptation Benchmark (VTAB) (Zhai et al., 2019), which spans 19 image classification datasets across natural, specialized, and structured domains. We use ViT-B/16 pretrained on ImageNet-22K as the backbone to ensure a consistent evaluation setup.

We compare FlexLoRA against strong baselines under a unified parameter budget. Specifically, we consider: (i) standard LoRA with a fixed-rank (Hu et al., 2021), and (ii) AdaLoRA (Zhang et al., 2023), both initialized with the same rank budget for fairness.

## 4.2 IMPLEMENTATION DETAILS

All methods are implemented in PyTorch (Paszke et al., 2019), based on Huggingface Transformers (Wolf et al., 2020). For FlexLoRA, we introduce a dynamic rank scheduler to stabilize rank adjustment. At training step $t$, the number of ranks adjusted, denoted $b(t)$, is defined by a cubic decay schedule:

$$b(t) = \text{round}\left(b_0 \cdot \left(1 - \frac{t - t_{\text{warmup}}}{T - t_{\text{final}}}\right)^3\right), \tag{4}$$

where $b_0$ is the initial adjustment size, $t_{\text{warmup}}$ marks the start of rank adaptation, $t_{\text{final}}$ denotes the beginning of the final freeze phase, and $T$ is the total number of training steps. We clamp $b(t)$ to the range $[0, b_0]$ and further restrict it so that adjustments never exceed the number of available modules or per-module rank limits. This schedule allows for aggressive rank reallocation in early training, when model capacity is being explored, and gradual stabilization towards convergence, ensuring consistent optimization in later stages. All methods are implemented in PyTorch (Paszke et al., 2019) with the HuggingFace Transformers library (Wolf et al., 2020). Experiments are conducted on NVIDIA A100 GPUs, and unless otherwise specified, hyper-parameters follow recommended settings in prior work to ensure fair comparison.

## 4.3 NATURAL LANGUAGE UNDERSTANDING

Table 1 reports the GLUE benchmark results, comparing FlexLoRA with LoRA ($r = 8$) and AdaLoRA under the same parameter budget. Details of the dataset and hyper-parameters settings can be found in Appendix D.1. FlexLoRA consistently delivers the best or competitive performance across all tasks. The gains are particularly pronounced on CoLA and RTE, where FlexLoRA significantly surpasses both baselines, highlighting its advantage in handling linguistically challenging tasks. Overall, FlexLoRA achieves the highest average score of 89.1, outperforming AdaLoRA (88.1) and LoRA (81.8), thereby demonstrating the effectiveness of entropy-guided, bidirectional rank allocation.

Table 1: Results on GLUE development set with DeBERTaV3-base. We report mean of 5 runs using different random seeds.

| Method | Params. | CoLA Mcc. | MNLI Acc. | MRPC Acc. | RTE Acc. | QNLI Acc. | SST-2 Acc. | STS-B Corr. | QQP Acc. | Avg. |
|---|---|---|---|---|---|---|---|---|---|---|
| Full FT | 184.3M | 69.2 | 89.9 | 90.2 | 83.8 | 94.0 | 95.6 | 91.6 | 92.4 | 88.3 |
| BitFit | 0.1M | $67.0_{\pm0.5}$ | $89.4_{\pm0.2}$ | $87.8_{\pm05}$ | $78.7_{\pm0.9}$ | $92.2_{\pm0.2}$ | $94.8_{\pm0.3}$ | $91.4_{\pm0.2}$ | $88.4_{\pm0.2}$ | 86.2 |
| H-Adapter | 1.2M | $62.6_{\pm3.2}$ | $86.5_{\pm0.4}$ | $89.9_{\pm2.3}$ | $80.4_{\pm2.0}$ | $92.8_{\pm0.2}$ | $93.7_{\pm0.4}$ | $90.2_{\pm1.1}$ | $90.8_{\pm0.1}$ | 85.9 |
| P-Adapter | 1.2M | $63.9_{\pm1.7}$ | $86.8_{\pm0.3}$ | $89.5_{\pm0.9}$ | $80.5_{\pm2.9}$ | $92.6_{\pm0.2}$ | $93.8_{\pm0.2}$ | $90.7_{\pm0.6}$ | $90.5_{\pm0.1}$ | 86.0 |
| AdapterFusion | 1.2M | $68.8_{\pm0.2}$ | $90.3_{\pm0.3}$ | $89.5_{\pm0.1}$ | $85.2_{\pm0.5}$ | $94.3_{\pm0.3}$ | $95.6_{\pm0.6}$ | $91.5_{\pm0.1}$ | $92.0_{\pm0.4}$ | 88.4 |
| LoRA$_{r=8}$ | 1.3M | $68.5_{\pm0.6}$ | $89.8_{\pm0.2}$ | $90.7_{\pm0.7}$ | $84.8_{\pm0.6}$ | $94.1_{\pm0.8}$ | $94.0_{\pm0.2}$ | $91.2_{\pm0.3}$ | $87.9_{\pm0.3}$ | 81.7 |
| AdaLoRA | 1.9M | $70.0_{\pm1.8}$ | $89.0_{\pm1.6}$ | $90.9_{\pm1.5}$ | $88.1_{\pm0.9}$ | $94.1_{\pm1.1}$ | $94.6_{\pm0.9}$ | $91.2_{\pm0.3}$ | $87.2_{\pm2.0}$ | 88.1 |
| DoRA | 1.3M | $65.4_{\pm0.4}$ | $87.8_{\pm0.2}$ | $90.1_{\pm0.3}$ | $81.7_{\pm1.8}$ | $93.0_{\pm0.0}$ | $91.3_{\pm0.2}$ | $91.3_{\pm0.0}$ | $91.3_{\pm0.5}$ | 86.5 |
| DyLoRA$_{r=8}$ | 0.9M | $59.5_{\pm1.0}$ | $86.8_{\pm0.1}$ | $91.4_{\pm0.8}$ | $77.6_{\pm0.6}$ | $93.0_{\pm0.3}$ | $94.4_{\pm0.4}$ | $91.1_{\pm0.2}$ | $89.9_{\pm0.1}$ | 85.5 |
| FlexLoRA | 1.9M | $71.8_{\pm0.9}$ | $90.0_{\pm0.7}$ | $90.9_{\pm0.6}$ | $88.8_{\pm0.7}$ | $94.2_{\pm0.2}$ | $95.2_{\pm0.4}$ | $91.5_{\pm0.1}$ | $90.3_{\pm0.7}$ | 89.1 |

## 4.4 COMMONSENSE REASONING

We next evaluate FlexLoRA on commonsense reasoning benchmarks. Details of the dataset and hyper-parameters settings can be found in Appendix D.2. Results in Table 2 show that FlexLoRA consistently delivers strong performance across all settings. On the LLaMA-3 models, FlexLoRA achieves clear improvements over standard LoRA, underscoring the necessity of adaptive rank allocation when operating under constrained parameter budgets. At rank 8, FlexLoRA achieves an

average score of 85.2 on LLaMA-3, slightly surpassing AdaLoRA (85.1). When the rank is increased to 32, FlexLoRA further improves to 85.5, establishing the best overall results among all parameter-efficient baselines, including LoRA-Dash, NoRA+, and PrecLoRA. These findings suggest that FlexLoRA not only adapts effectively to earlier model generations but also remains competitive on the latest LLaMA-3, demonstrating robustness across model iterations while exploiting higher ranks without redundancy.

Table 2: Results on commonsense reasoning tasks.

| Model | Method | Param. | BoolQ | PIQA | SIQA | HellaS. | WinoG. | ARC-e | ARC-c | OBQA | Avg. |
|---|---|---|---|---|---|---|---|---|---|---|---|
| ChatGPT | - | - | 73.1 | 85.4 | 68.5 | 78.5 | 66.1 | 89.8 | 79.9 | 74.8 | 77.0 |
| LLaMA3-8B | Full FT | 8B | 75.3 | 89.9 | 81.5 | 95.8 | 87.6 | 91.6 | 79.3 | 87.4 | 86.1 |
| | $LoRA_{r=8}$ | 14.2M | 62.2 | 86.5 | 80.3 | 94.5 | 84.4 | 77.7 | 88.0 | 85.8 | 82.4 |
| | $AdaLoRA_{r=8}$ | 21.2M | 74.1 | 88.4 | 80.3 | 95.7 | 84.4 | 80.1 | 91.0 | 87.0 | 85.1 |
| | $FlexLoRA_{r=8}$ | 21.2M | 74.3 | 88.6 | 81.0 | 95.6 | 84.9 | 80.2 | 90.7 | 86.4 | 85.2 |
| | $LoRA_{r=32}$ | 56.6M | 75.6 | 89.5 | 81.2 | 95.1 | 85.1 | 80.1 | 90.3 | 86.2 | 84.5 |
| | $AdaLoRA_{r=32}$ | 56.6M | 71.3 | 88.7 | 80.1 | 94.5 | 86.2 | 78.8 | 90.2 | 85.8 | 84.5 |
| | LoRA-Dash | 56.6M | 75.3 | 88.5 | 80.2 | 95.7 | 86.8 | 90.7 | 80.2 | 85.6 | 85.4 |
| | NoRA+ | 56.6M | 71.2 | 85.1 | 79.5 | 92.2 | 83.4 | 85.9 | 72.3 | 83.2 | 81.6 |
| | PrecLoRA | 56.6M | 70.7 | 85.8 | 78.9 | 91.9 | 83.7 | 85.1 | 71.1 | 82.4 | 81.2 |
| | $FlexLoRA_{r=32}$ | 56.6M | 72.8 | 89.1 | 80.7 | 96.0 | 86.4 | 81.3 | 90.8 | 87.2 | 85.5 |

## 4.5 VISUAL TASK

To further assess the generality of FlexLoRA beyond NLP, we evaluate it on the Visual Task Adaptation Benchmark (VTAB) and compare against LoRA ($r = 14$) and AdaLoRA under identical experimental conditions. Details of the dataset and hyper-parameters settings can be found in Appendix D.3. As shown in Table 3, FlexLoRA achieves the highest average accuracy of 67.8%, outperforming both LoRA (66.7%) and AdaLoRA (64.7%) under comparable parameter budgets. Notably, FlexLoRA yields substantial improvements on natural image datasets (e.g., +8.9 on CIFAR100 over LoRA), highlighting its ability to capture domain diversity. It also performs strongly on specialized datasets such as Camelyon and Retinopathy, demonstrating robustness in medical and fine-grained visual recognition. Overall, these results confirm that FlexLoRA's entropy-guided rank allocation is not confined to language but generalizes effectively to vision.

Table 3: Results on VTAB benchmark. Accuracy (%) across Natural, Specialized, and Structured domains.

| Method | Param. | Natural | | | | | | | Specialized | | | | Structured | | | | | | | | Avg. |
|---|---|---|---|---|---|---|---|---|---|---|---|---|---|---|---|---|---|---|---|---|---|
| | | Cifar100 | Caltech101 | DTD | Flower102 | Pets | SVHN | Sun397 | Camelyon | EuroSAT | Resisc45 | Retinopathy | Clevr-Count | Clevr-Dist | DMLab | KITTI-Dist | dSpr-Loc | dSpr-Ori | sNORB-Azim | sNORB-Ele | |
| Full FT | 327M | 68.9 | 87.7 | 64.3 | 97.2 | 86.9 | 84.7 | 38.8 | 79.7 | 95.7 | 84.2 | 73.9 | 56.3 | 58.6 | 41.7 | 65.5 | 57.5 | 46.7 | 25.7 | 29.1 | 68.9 |
| $LoRA_{r=14}$ | 1.29M | 49.3 | 87.7 | 63.1 | 98.1 | 87.5 | 73.2 | 47.0 | 79.4 | 94.4 | 80.1 | 71.5 | 74.3 | 60.4 | 40.4 | 72.2 | 67.9 | 40.8 | 13.7 | 29.3 | 66.7 |
| FLoRA | 1.11M | 50.5 | 86.5 | 63.9 | 98.1 | 87.4 | 72.0 | 49.4 | 78.6 | 94.2 | 78.3 | 74.0 | 67.4 | 57.2 | 40.4 | 74.4 | 65.9 | 38.0 | 12.0 | 30.1 | 66.4 |
| LieRA | 1.11M | 52.3 | 86.1 | 65.8 | 98.4 | 89.3 | 57.7 | 50.4 | 78.6 | 91.6 | 76.1 | 72.3 | 57.7 | 49.5 | 38.0 | 72.7 | 60.0 | 36.1 | 12.3 | 26.7 | 64.4 |
| MiLoRA | 1.11M | 39.3 | 83.6 | 61.6 | 97.0 | 86.2 | 33.4 | 48.1 | 77.0 | 85.9 | 66.8 | 73.9 | 31.5 | 29.7 | 34.0 | 54.0 | 12.0 | 16.7 | 10.9 | 21.2 | 55.2 |
| PiSSA | 1.11M | 48.8 | 87.3 | 62.9 | 98.0 | 88.2 | 67.8 | 47.2 | 80.5 | 94.6 | 79.2 | 70.4 | 73.2 | 54.2 | 41.7 | 74.3 | 70.7 | 40.9 | 12.7 | 29.8 | 66.3 |
| MoSLoRA | 1.11M | 48.0 | 85.5 | 60.5 | 97.7 | 86.0 | 74.0 | 47.7 | 77.6 | 94.2 | 76.6 | 74.2 | 74.2 | 59.9 | 41.1 | 74.4 | 66.6 | 37.5 | 13.1 | 30.2 | 66.0 |
| AdaLoRA | 1.26M | 56.9 | 87.2 | 63.3 | 98.4 | 88.3 | 69.5 | 51.2 | 76.4 | 92.6 | 76.5 | 72.3 | 56.5 | 51.8 | 39.5 | 75.3 | 45.1 | 35.9 | 12.7 | 27.1 | 64.7 |
| MLAE | 1.11M | 49.0 | 87.1 | 62.4 | 97.9 | 88.1 | 69.8 | 47.6 | 79.9 | 94.4 | 80 | 70.1 | 73.5 | 57.4 | 42.3 | 74 | 69.2 | 43.2 | 13.2 | 30.2 | 66.6 |
| FlexLoRA | 1.18M | 58.2 | 88.9 | 64.3 | 98.5 | 89.3 | 73.7 | 51.3 | 80.0 | 93.4 | 80.3 | 73.4 | 66.9 | 60.0 | 41.0 | 77.8 | 64.7 | 39.0 | 13.6 | 29.9 | 67.8 |

## 5 FURTHER STUDY

To better understand the design choices of FlexLoRA, we conduct a series of ablation studies and exploratory analyses. We compare alternative importance metrics, examine the necessity of bidirectional rank allocation, evaluate different initialization strategies, and analyze the final distribution of ranks after training. Together, these studies shed light on why FlexLoRA works and provide insights into future research directions.

## 5.1 IMPORTANCE METRICS

We first investigate the role of importance metrics in guiding rank allocation. In addition to our proposed entropy-based criterion, we compare two widely adopted alternatives: sensitivity-based and norm-based metrics.

**Sensitivity-based metrics.** As in AdaLoRA (Sec. 2.2), importance can be estimated by element-level gradient–weight products:

$$I(w_{ij}) = |w_{ij} \cdot \nabla_{w_{ij}} \mathcal{L}|, \tag{5}$$

which approximates the sensitivity of each parameter to the training loss. To reduce the noise and instability of such estimates, AdaLoRA further applies exponential moving average smoothing and an uncertainty term to obtain a refined importance score:

$$s^{(t)}(w_{ij}) = \bar{I}^{(t)}(w_{ij}) \cdot \bar{U}^{(t)}(w_{ij}), \tag{6}$$

where $\bar{I}^{(t)}$ is the smoothed sensitivity and $\bar{U}^{(t)}$ quantifies local variation. Although intuitive, such heuristics are highly sensitive to gradient noise, unstable across iterations, and fail to incorporate matrix-level structural information.

**Norm-based metrics.** The nuclear norm and Frobenius norm summarize the overall strength of singular values by aggregating spectral magnitudes. While these metrics capture cumulative energy, they cannot reflect the distributional structure of singular components. For consistency, both norms are normalized by the number of singular values $n$:

$$I_{\text{nuclear}} = \frac{1}{n} \sum_{i=1}^{n} |\lambda_i|, \quad I_{\text{F}} = \frac{1}{n} \sqrt{\sum_{i=1}^{n} \lambda_i^2}. \tag{7}$$

**Results.** Table 4 shows that entropy consistently outperforms both sensitivity- and norm-based alternatives on representative GLUE tasks. These results highlight entropy's superior discriminability and robustness, confirming it as a more principled criterion for dynamic rank allocation.

Table 4: Comparison of different importance metrics on GLUE with DeBERTaV3-base. Entropy consistently outperforms the other norm-based alternatives.

| Method | Params. | CoLA Mcc. | MNLI Acc. | MRPC Acc. | RTE Acc. | QNLI Acc. | SST-2 Acc. | STS-B Corr. | QQP Acc. | Avg. |
|--------|---------|-----------|-----------|-----------|----------|-----------|------------|-------------|----------|------|
| Full FT | 184.3M | 69.2 | 89.9 | 90.2 | 83.8 | 94.0 | 95.6 | 91.6 | 92.4 | 88.3 |
| AdaLoRA | 1.9M | 70.0 | 89.0 | 90.9 | 88.1 | 94.1 | 94.6 | 91.2 | 87.2 | 88.1 |
| Nuclear | 1.9M | 69.8 | 89.1 | 90.2 | 84.8 | 94.4 | 94.8 | 91.1 | 87.5 | 87.7 |
| Frobenius | 1.9M | 69.4 | 88.8 | 86.5 | 86.3 | 94.1 | 94.7 | 90.8 | 86.4 | 87.1 |
| FlexLoRA | 1.9M | 71.8 | 90.0 | 90.9 | 88.8 | 94.2 | 95.2 | 91.5 | 90.3 | 89.1 |

## 5.2 RANK PRUNING AND EXPANSION

We next assess the necessity of combining pruning and expansion. Specifically, we compare FlexLoRA with two variants: (i) **Prune-only**, which removes low-importance singular directions but never expands capacity; and (ii) **Expand-only**, which continually adds directions without pruning.

As shown in Table 5, prune-only leads to over-pruning and lacks the flexibility to recover capacity, while expand-only wastes parameters by retaining redundant directions. In contrast, FlexLoRA jointly prunes and expands ranks under a global budget, achieving consistently superior results. This confirms that bidirectional rank adjustment is critical for balancing efficiency and adaptability, and motivates further exploration of adaptive scheduling strategies.

## 5.3 ZERO-IMPACT INITIALIZATION

We further analyze initialization strategies for newly added singular directions.

Table 5: Results of further study on pruning and expansion. We report results on four representative GLUE benchmarks (CoLA, MRPC, RTE, and STS-B) using DeBERTaV3-base. Results show that FlexLoRA's pruning and expansion strategy outperforms prune-only and expand-only strategies.

| Method | Params. | CoLA Mcc. | MNLI Acc. | MRPC Acc. | RTE Acc. | QNLI Acc. | SST-2 Acc. | STS-B Corr. | QQP Acc. | Avg. |
|---|---|---|---|---|---|---|---|---|---|---|
| Full FT | 184.3M | 69.2 | 89.9 | 90.2 | 83.8 | 94.0 | 95.6 | 91.6 | 92.4 | 88.3 |
| Prune-only | 1.9M | 66.8 | 89.4 | 90.4 | 85.9 | 94.2 | 94.5 | 91.4 | 87.6 | 87.5 |
| Expand-only | 1.9M | 68.3 | 89.5 | 89.5 | 87.7 | 94.3 | 93.8 | 91.3 | 86.5 | 87.6 |
| FlexLoRA | 1.9M | 71.8 | 90.0 | 90.9 | 88.8 | 94.2 | 95.2 | 91.5 | 90.3 | 89.1 |

**Zero-impact initialization**, adopted in FlexLoRA, sets the singular value to zero while sampling vectors from a Gaussian distribution. As alternatives, we examine: (i) **Small-init**, which assigns small non-zero values to new singular values and samples orthogonal vectors via Gram–Schmidt; (ii) **Zero-init**, which sets both values and vectors to zero, freezing new directions until gradients accumulate; (iii) **Orthogonal-init**, which sets values to zero but samples orthogonal vectors to preserve independence.

Table 6 shows that FlexLoRA with zero-impact initialization achieves the highest average score (85.8), consistently outperforming all alternative initialization strategies across the four GLUE benchmarks. This demonstrates that zero-impact initialization provides the most favorable balance between stability and learnability, ensuring that the expanded capacity is fully utilized to enhance task-specific adaptation.

Table 6: Results of further study on zero-impact initialization. We report results on four representative GLUE benchmarks (CoLA, MRPC, RTE, and STS-B) using DeBERTaV3-base. Results show that FlexLoRA's zero-impact initialization achieves the best balance, outperforming the other strategies.

| Method | Params. | CoLA Mcc. | MNLI Acc. | MRPC Acc. | RTE Acc. | QNLI Acc. | SST-2 Acc. | STS-B Corr. | QQP Acc. | Avg. |
|---|---|---|---|---|---|---|---|---|---|---|
| Full FT | 184.3M | 69.2 | 89.9 | 90.2 | 83.8 | 94.0 | 95.6 | 91.6 | 92.4 | 88.3 |
| Small-init | 1.9M | 69.2 | 89.4 | 89.7 | 85.9 | 94.4 | 95.2 | 91.2 | 87.5 | 87.8 |
| Zero-init | 1.9M | 66.4 | 89.5 | 88.2 | 84.8 | 94.4 | 94.6 | 91.1 | 87.4 | 87.1 |
| Orthogonal-init | 1.9M | 70.4 | 89.8 | 90.0 | 87.0 | 94.2 | 94.7 | 90.7 | 87.5 | 88.0 |
| FlexLoRA | 1.9M | 71.8 | 90.0 | 90.9 | 88.8 | 94.2 | 95.2 | 91.5 | 90.3 | 89.1 |

## 5.4 RANK DISTRIBUTION AFTER ALLOCATION

Finally, we analyze how FlexLoRA allocates ranks across layers and modules during training. Figure 2 visualizes the evolution of effective ranks on the CoLA task, where darker colors denote higher capacity allocation and lighter colors indicate stronger pruning.

The results reveal several interesting patterns. Contrary to the common assumption that deeper layers should dominate task-specific adaptation, we observe that shallower layers (L0–L3) undergo substantial rank expansion. This indicates that early layers capture task-relevant syntactic and semantic cues that require increased modeling capacity, highlighting their underestimated role in linguistic adaptation. In contrast, deeper layers (L9–L11) are consistently pruned across training, suggesting that these layers encode more task-agnostic or redundant representations that can be preserved in compressed form without harming performance. Middle layers (L4–L8) exhibit mixed behavior. Within these layers, some modules maintain moderate rank allocations, while others are gradually pruned depending on their relative contribution to the task. Notably, attention output and intermediate dense modules in these layers tend to preserve higher ranks, suggesting that they play a central role in shaping task-specific decision boundaries and information propagation.

Overall, these findings suggest that CoLA-relevant signals are primarily concentrated in shallow and middle layers, while deeper layers encode more generic features requiring less adaptation. This

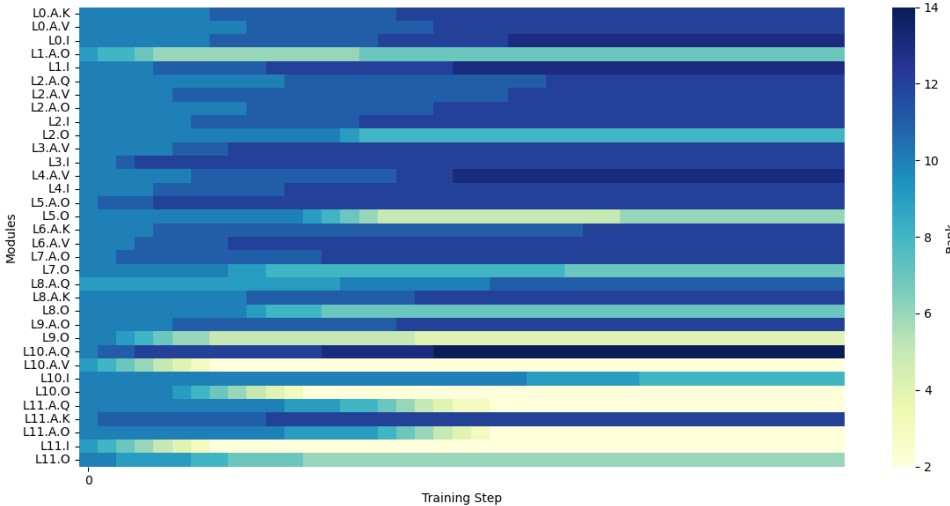

Figure 2: Visualization of rank allocation during FlexLoRA training on the CoLA task. We selected the modules with the most significant changes in rank and sorted them by layer depth. Modules are sorted by layer depth from top (shallow) to bottom (deep). K, V, Q denote key, value, and query projections; A is attention output; I is intermediate dense; O is output dense. Darker colors indicate more capacity, lighter colors indicate stronger pruning.

demonstrates that FlexLoRA adaptively reallocates capacity in a manner that aligns with linguistic task requirements, improving efficiency while also yields interpretable rank allocation patterns.

## 6 CONCLUSION

We presented FlexLoRA, a flexible low-rank adaptation framework that addresses the limitations of existing PEFT methods. FlexLoRA introduces three synergistic components: a matrix-level entropy-guided importance metric, a bidirectional rank allocation mechanism under a global budget, and a zero-impact initialization strategy. These components together ensure both stable optimization and efficient utilization of model capacity during fine-tuning. Extensive experiments on natural language understanding, commonsense reasoning, and visual recognition benchmarks demonstrate that FlexLoRA consistently outperforms strong state-of-the-art baselines in both accuracy and parameter efficiency. Further analyses confirm the effectiveness of its core components and reveal interpretable rank distribution patterns that aligned with task-specific requirements, offering insights into the functional roles of different layers and modules. In summary, FlexLoRA provides a robust and general strategy for parameter-efficient fine-tuning, laying a principled foundation for future research on flexible low-rank adaptation of large pre-trained models across diverse modalities and tasks.

### ACKNOWLEDGMENTS

This work was supported by the National Natural Science Foundation of China under Grant U24A20322 and Grant 62576094.

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

## A    THE USE OF LLM

We used a large language model (ChatGPT) as a general-purpose writing assistant. Its role was limited to the following:

- **Language polishing:** Refining grammar, improving clarity, and ensuring consistent academic style in certain parts of the paper.
- **Formatting suggestions:** Providing alternative sentence structures or paragraph organization for better readability.

No part of the research design, methodology, experiments, or analysis was generated by the language model. All technical content, scientific claims, and conclusions are the sole work and responsibility of the authors.

The use of LLMs did not rise to the level of substantive contribution that would merit authorship.

## B    NORMALIZATION OF SPECTRAL ENTROPY

Recall the spectral entropy for normalized squared singular values

$$s_i = \frac{\lambda_i^2}{\sum_{j=1}^r \lambda_j^2}, \qquad s_i > 0, \ \sum_{i=1}^r s_i = 1,$$

is given by

$$H(s) = -\sum_{i=1}^r s_i \log s_i,$$

defined on the probability simplex $\Delta^{r-1} = \{s \in \mathbb{R}^r : s_i > 0, \ \sum_i s_i = 1\}$.

Maximize $H(s)$ subject to $\sum_i s_i = 1$. Form the Lagrangian

$$\mathcal{L}(s, \mu) = -\sum_{i=1}^r s_i \log s_i + \mu\Big(\sum_{i=1}^r s_i - 1\Big).$$

Taking partial derivatives and setting them to zero gives, for each $i$,

$$\frac{\partial \mathcal{L}}{\partial s_i} = -(\log s_i + 1) + \mu = 0 \quad \implies \quad \log s_i = \mu - 1.$$

Hence $s_i = e^{\mu-1}$ is constant for all $i$. Using the constraint $\sum_i s_i = 1$ yields

$$re^{\mu-1} = 1 \quad \implies \quad e^{\mu-1} = \frac{1}{r},$$

and therefore

$$s_i = \frac{1}{r} \qquad \text{for all } i.$$

Substituting the uniform distribution into $H$ gives

$$H_{\max} = H\left(\tfrac{1}{r}, \ldots, \tfrac{1}{r}\right) = -\sum_{i=1}^{r} \frac{1}{r} \log \frac{1}{r} = \log r.$$

To show this stationary point is the global maximum, note that $H(s)$ is strictly concave on $\Delta^{r-1}$. Indeed, the Hessian matrix of $H$ (with respect to the coordinates $s_i$) is diagonal:

$$\frac{\partial^2 H}{\partial s_i^2} = -\frac{1}{s_i}, \qquad \frac{\partial^2 H}{\partial s_i \partial s_j} = 0 \; (i \neq j).$$

For any nonzero vector $x \in \mathbb{R}^r$,

$$x^\top \nabla^2 H \, x = -\sum_{i=1}^{r} \frac{x_i^2}{s_i} < 0$$

because $s_i > 0$. Thus $\nabla^2 H$ is negative definite on the domain, implying $H$ is strictly concave. A strictly concave function on a convex set has at most one stationary point and that stationary point is the global maximum. Therefore the uniform distribution $s_i = 1/r$ yields the unique global maximum and $H_{\max} = \log r$.

Dividing $H(s)$ by $\log r$ yields a normalized score

$$I(\mathbf{\Lambda}) = \frac{H(\mathbf{\Lambda})}{\log r} \in [0, 1],$$

with $I = 1$ at the uniform distribution and $I \to 0$ when the distribution concentrates its mass on a single component.

## C    MONOTONICITY OF IMPORTANCE FOR A SINGLE SINGULAR VALUE

Given a matrix with singular values $\lambda_1, \ldots, \lambda_r$, recall that the spectral importance of the $i$-th singular value is

$$s_i = \frac{\lambda_i^2}{\sum_{j=1}^{r} \lambda_j^2}, \quad I(\mathbf{\Lambda}) = -\sum_{i=1}^{r} s_i \log(s_i + \epsilon), \tag{8}$$

where $\epsilon > 0$ is a small constant to avoid numerical issues.

To see that $I(\lambda_i)$ is monotonic in $\lambda_i$, consider $s_i$ as a function of $\lambda_i$:

$$\frac{\partial s_i}{\partial \lambda_i} = \frac{2\lambda_i(\sum_{j \neq i} \lambda_j^2)}{(\sum_{j=1}^{r} \lambda_j^2)^2} > 0 \quad \text{for } \lambda_i > 0. \tag{9}$$

Since $-s_i \log s_i$ is an increasing function for small $s_i \in (0, 1)$, we have that decreasing $\lambda_i$ decreases $s_i$, which in turn decreases its contribution to the total entropy $I(\mathbf{\Lambda})$.

Hence, the smallest singular value $\lambda_{\min}$ always corresponds to the smallest importance $I(\lambda_{\min})$, justifying the pruning strategy in Sec 3.3.

## D    DETAILS ON EXPERIMENTS

### D.1    DETAILS ON NATURAL LANGUAGE UNDERSTANDING TASK

For the natural language understanding (NLU) experiments, we adopt the General Language Understanding Evaluation (GLUE) Wang et al. (2018) benchmark, a suite of tasks designed to assess a model's broad linguistic competence. GLUE comprises two single-sentence classification tasks,

CoLA Warstadt et al. (2019) and SST-2 Socher et al. (2013), three tasks focused on semantic similarity and paraphrase detection, MRPC Dolan & Brockett (2005), QQP Wang et al. (2018), and STS-B Cer et al. (2017), and three natural language inference tasks, MNLI Williams et al. (2017), QNLI Rajpurkar et al. (2016), and RTE Dagan et al. (2005); Bar-Haim et al. (2006); Giampiccolo et al. (2007); Bentivogli et al. (2009). The details of these datasets are shown in Table 7.

For this benchmark, we fine-tune DeBERTaV3-base He et al. (2021b) models. The hyper-parameter settings for this task is shown in Table 8.

Table 7: Details of GLUE dataset. Combined Score for MRPC and QQP is defined as the average of Accuracy and F1, while for STS-B it is the average of Pearson and Spearman correlations

| Dataset | Task | # Train | # Dev | # Test | # Label | Metrics |
|---------|------|---------|-------|--------|---------|---------|
| Single-Sentence Classification | | | | | | |
| CoLA | Acceptability | 8.5k | 1k | 1k | 2 | Matthews corr |
| SST-2 | Sentiment | 67k | 872 | 1.8k | 2 | Accuracy |
| Similarity and Paraphrase | | | | | | |
| MRPC | Paraphrase | 3.7k | 408 | 1.7k | 2 | Combined Score |
| QQP | Paraphrase | 364k | 40k | 391k | 2 | Combined Score |
| STS-B | Similarity | 7k | 1.5k | 1.4k | 1 | Combined Score |
| Natural Language Inference | | | | | | |
| MNLI | NLI | 393k | 20k | 20k | 3 | Accuracy |
| QNLI | QA/NLI | 108k | 5.7k | 5.7k | 2 | Accuracy |
| RTE | NLI | 2.5k | 276 | 3k | 2 | Accuracy |

Table 8: Hyper-parameter settings of FlexLoRA on NLU task.

| Hyper-parameter | CoLA | MNLI | MRPC | RTE | QNLI | SST-2 | STS-B | QQP |
|-----------------|------|------|------|-----|------|-------|-------|-----|
| Optimizer | AdamW | | | | | | | |
| Warmup Ratio | 0.1 | | | | | | | |
| LR schedule | Linear | | | | | | | |
| Rank $r$ | 8 | | | | | | | |
| $\alpha$ | 16 | | | | | | | |
| $b$ | 4 | | | | | | | |
| Max Seq. Len. | 64 | 256 | 256 | 512 | 256 | 256 | 256 | 320 |
| Batch Size | 32 | 32 | 32 | 32 | 32 | 32 | 32 | 32 |
| Learning Rate | 8e-4 | 5e-4 | 1e-3 | 1.2e-3 | 5e-4 | 8e-4 | 2.2e-3 | 8e-4 |
| Epochs | 20 | 12 | 30 | 50 | 5 | 20 | 20 | 5 |
| $T_{warmup}$ | 1000 | 5000 | 500 | 500 | 1000 | 5000 | 1000 | 5000 |
| $T_{final}$ | 1000 | 5000 | 500 | 500 | 1000 | 5000 | 1000 | 5000 |
| $\Delta_T$ | 200 | 1000 | 100 | 100 | 200 | 1000 | 200 | 1000 |

## D.2 DETAILS ON COMMONSENSE REASONING TASK

The commonsense reasoning benchmark suite comprises eight sub-tasks, each associated with a specific dataset: BoolQ Clark et al. (2019), PIQA Bisk et al. (2020), SIQA Sap et al. (2019), HellaS. Zellers et al. (2019), WinoG. Sakaguchi et al. (2021), ARC-e/ARC-c Clark et al. (2018), OBQA Mihaylov et al. (2018). Following the protocol described in Hu et al. (2023b), we aggregate the training portions of all tasks into a unified corpus, referred to as the Commonsense170K dataset, and then evaluate performance separately on each task's test set. The hyper-parameter settings of FlexLoRA are shown in Table. 9.

We fine-tune LLaMA3-8B AI@Meta (2024) on this task. For comparison, we also include results from ChatGPT's implementation with the gpt-3.5-turbo API, particularly focusing on zero-shot Chain of Thought approaches Wei et al. (2022). The results of fully fine-tuning(Full FT) and ChatGPT are cited from Liu et al. (2024).

Table 9: Hyper-parameter settings of FlexLoRA on commonsense reasoning task.

| Hyper-parameters | LLaMA3-8B | Hyper-parameters | LLaMA3-8B |
|---|---|---|---|
| Rank $r$ | 8 & 32 | $\alpha$ | 16 |
| Learning Rate | 3e-4 | LR Scheduler | Linear |
| Dropout | 0.05 | Optimizer | AdamW |
| Batch size | 16 | Warmup Steps | 100 |
| Epochs | 3 | $b$ | 4 |
| $T_{warmup}$ | 5000 | $T_{final}$ | 5000 |
| $\Delta_T$ | 1000 | Where | Q, K, V, Up, Down |

## D.3 DETAILS ON VISUAL TASK

As shown in Table 10, the VTAB-1K dataset (Zhai et al., 2019) allocates 800 samples for training and 200 for validation during hyper-parameter tuning. The final model is trained on all 1,000 samples and evaluated on the official test set.

For this benchmark, we fine-tune ViT-B/16(Dosovitskiy, 2020) models. The results of fully fine-tuning(Full FT) are cited from Jie & Deng (2023). The hyper-parameter settings for this task is shown in Table 11.

Table 10: Details of VTAB-1K dataset.

| Dataset | Natural | | | | | | | Specialized | | | | Structured | | | | | | | |
|---|---|---|---|---|---|---|---|---|---|---|---|---|---|---|---|---|---|---|---|
| | Cifar100 | Caltech101 | DTD | Flower102 | Pets | SVHN | Sun397 | Camelyon | EuroSAT | Resisc45 | Retinopathy | Clevr-Count | Clevr-Dist | DMLab | KITTI-Dist | dSpr-Loc | dSpr-Ori | sNORB-Azim | sNORB-Ele |
| # Classes | 100 | 102 | 47 | 102 | 37 | 10 | 397 | 2 | 10 | 45 | 5 | 8 | 6 | 6 | 4 | 16 | 16 | 18 | 18 |
| Train | 800/1000 | | | | | | | | | | | | | | | | | | |
| Val | 200 | | | | | | | | | | | | | | | | | | |
| Test | 10000 | 6084 | 1880 | 6149 | 3669 | 26032 | 21750 | 32768 | 5400 | 6300 | 42670 | 15000 | 15000 | 22735 | 711 | 73728 | 73728 | 12150 | 12150 |

## E COMPARISON OF TRAINING COST

In this part, we report the system-level metrics, including GPU memory and train runtime. As shown in Table 12-14, the results consistently show that FlexLoRA does not introduce noticeable overhead compared with AdaLoRA across all benchmarks. GPU memory usage remains nearly

Table 11: Hyper-parameter settings of FlexLoRA on visual task.

| Optimizer | batch size | Learning Rate | Epochs | $b$ | $T_{warmup}$ | $T_{final}$ | $\Delta_T$ |
|-----------|------------|---------------|--------|-----|--------------|-------------|------------|
| AdamW | 32 | 1e-3 | 100 | 4 | 500 | 500 | 100 |

identical across all settings, indicating that the rank reallocation strategy in FlexLoRA brings small computational cost.

As shown in Table 12, for NLU tasks (CoLA, RTE), FlexLoRA matches AdaLoRA in both runtime and memory while outperforming it in evaluation speed, and it incurs only small runtime differences compared with LoRA. As shown in Table 13, FlexLoRA even reduces memory usage and shortens training time relative to AdaLoRA on commonsense reasoning with LLaMA3-8B. As shown in Table 14, for visual tasks (Cifar100, Resisc45, DMLab), FlexLoRA again shows almost the same training cost as AdaLoRA, with runtime differences typically under 1%, while maintaining stable FLOP and memory footprints.

Table 12: Comparison of training cost on NLU tasks.

| Dataset | Metric | FlexLoRA | AdaLoRA | LoRA |
|---------|--------|----------|---------|------|
| CoLA | Train Runtime (s) | 2162.6 | 2167.9 | 2117.9 |
| | GPU Mem (MB) | 17446 | 17487 | 17443 |
| | Total FLOPs | $4.88 \times 10^{16}$ | $4.90 \times 10^{16}$ | $4.88 \times 10^{16}$ |
| | Eval Runtime (s) | 9.3231 | 9.2842 | 10.4523 |
| RTE | Train Runtime (s) | 1597.4 | 1574.9 | 1534.5 |
| | GPU Mem (MB) | 17446 | 17487 | 17443 |
| | Total FLOPs | $3.55 \times 10^{16}$ | $3.56 \times 10^{16}$ | $3.55 \times 10^{16}$ |
| | Eval Runtime (s) | 2.4398 | 2.3954 | 2.6691 |

Table 13: Comparison of training cost on commonsense reasoning tasks.

| Model | Metric | FlexLoRA | AdaLoRA | LoRA | LoRA-Dash |
|-------|--------|----------|---------|------|-----------|
| LLaMA3-8B | Train Runtime (h) | 6.01 | 6.47 | 4.76 | 6.77 |
| | GPU Mem (GB) | 38.0 | 43.3 | 37.6 | 51.8 |

## F  SINGULAR VALUE DISTRIBUTION

To further examine the rank allocation behavior, we randomly sampled ten LoRA modules trained on the CoLA task and visualized their singular value distributions. As shown in the Figure 3, we can notice that no matrix with small magnitude and high entropy that has obtained a large number of rank values, indicating that the magnitude of the singular values has only a minor effect on our rank allocation process, supporting the robustness of our adaptive strategy.

## G  MORE STUDY ON IMPORTANCE METRICS

To further validate the robustness of the spectral entropy metric and investigate the necessity of incorporating spectral energy into the importance metric, we designed and evaluated two hybrid variants that combine energy with entropy:

Table 14: Comparison of training cost on visual tasks.

| Dataset | Metric | FlexLoRA | AdaLoRA | LoRA |
|---|---|---|---|---|
| Cifar100 | Train Runtime (s) | 650 | 654 | 581 |
| | GPU Mem (MB) | 8414 | 8467 | 7787 |
| | Total FLOPs | $5.62 \times 10^{18}$ | $5.64 \times 10^{18}$ | $5.63 \times 10^{18}$ |
| | Eval Runtime (s) | 29.94 | 30.22 | 27.18 |
| Resisc45 | Train Runtime (s) | 646 | 647 | 566 |
| | GPU Mem (MB) | 8439 | 8449 | 7785 |
| | Total FLOPs | $5.62 \times 10^{18}$ | $5.64 \times 10^{18}$ | $5.63 \times 10^{18}$ |
| | Eval Runtime (s) | 19.74 | 19.78 | 17.82 |
| DMLab | Train Runtime (s) | 695 | 699 | 774 |
| | GPU Mem (MB) | 8451 | 8465 | 7785 |
| | Total FLOPs | $5.62 \times 10^{18}$ | $5.64 \times 10^{18}$ | $5.63 \times 10^{18}$ |
| | Eval Runtime (s) | 67.06 | 66.98 | 59.91 |

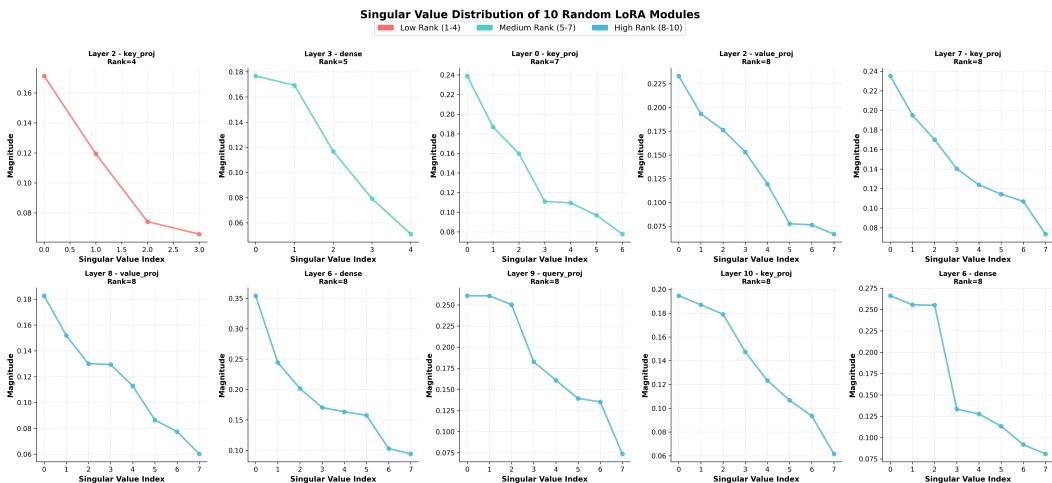

Figure 3: Singular value distributions of ten randomly sampled LoRA modules trained on the CoLA task.

**Elem-wise Energy Entropy** ($I_{Elem}$)**:** This variant calculates entropy at the element level within the singular value distribution, defined as:

$$I_{Elem} = -\frac{1}{r \log r} \sum_i \lambda_i s_i \log(s_i + \epsilon), \quad s_i = \frac{\lambda_i^2}{\sum_j \lambda_j^2}. \tag{10}$$

**Matrix-wise Energy Entropy** ($I_{Mat}$)**:** This variant aggregates the singular values before computing the entropy-based importance:

$$I_{Mat} = -\frac{1}{r \log r} \left( \sum_i \lambda_i \right) \left( \sum_i s_i \log(s_i + \epsilon) \right), \quad s_i = \frac{\lambda_i^2}{\sum_j \lambda_j^2}. \tag{11}$$

As shown in Table 15, neither variant outperforms the original FlexLoRA, reinforcing that our spectral energy entropy is a more reliable metric for measuring matrix importance.

Table 15: Comparison of FlexLoRA and two energy-considering variants on NLU tasks using DeBERTaV3-base.

| Method | Params. | CoLA Mcc. | MNLI Acc. | MRPC Acc. | RTE Acc. | QNLI Acc. | SST-2 Acc. | STS-B Corr. | QQP Acc. | Avg. |
|---|---|---|---|---|---|---|---|---|---|---|
| Full FT | 184.3M | 69.2 | 89.9 | 90.2 | 83.8 | 94.0 | 95.6 | 91.6 | 92.4 | 88.3 |
| Elem | 1.9M | 68.1 | 89.1 | 91.0 | 86.3 | 94.5 | 94.7 | 90.9 | 87.4 | 87.8 |
| Mat | 1.9M | 69.0 | 89.5 | 89.3 | 83.8 | 94.6 | 95.5 | 91.8 | 87.2 | 87.6 |
| FlexLoRA | 1.9M | 71.8 | 90.0 | 90.9 | 88.8 | 94.2 | 95.2 | 91.5 | 90.3 | 89.1 |

# H  MORE EXPERIMENTS ON VISUAL TASK

To further verify the effectiveness of our method on visual tasks, we conducted experiments following the settings in Xin et al. (2024)(learning rate = 2e-3, weight decay = 1e-3). Results in Table 16 demonstrate that FlexLoRA achieves a superior average accuracy of 76.0%, outperforming LoRA (74.5%) and AdaLoRA (75.1%) under comparable parameter budgets. These findings confirm that FlexLoRA's entropy-guided rank allocation continually maintains consistent efficacy and robustness when extended to visual tasks.

Table 16: Results on VTAB benchmark with the experimental settings in Xin et al. (2024). Accuracy (%) across Natural, Specialized, and Structured domains.

| Method | Param.(M) | Natural | | | | | | | Specialized | | | | Structured | | | | | | | | Avg. |
|---|---|---|---|---|---|---|---|---|---|---|---|---|---|---|---|---|---|---|---|---|---|
| | | Cifar100 | Caltech101 | DTD | Flower102 | Pets | SVHN | Sun397 | Camelyon | EuroSAT | Resisc45 | Retinopathy | Clevr-Count | Clevr-Dist | DMLab | KITTI-Dist | dSpr-Loc | dSpr-Ori | sNORB-Azim | sNORB-Ele | |
| Full FT | 85.8 | 68.9 | 87.7 | 64.3 | 97.2 | 86.9 | 84.7 | 38.8 | 79.7 | 95.7 | 84.2 | 73.9 | 56.3 | 58.6 | 41.7 | 65.5 | 57.5 | 46.7 | 25.7 | 29.1 | 68.9 |
| Linear | 0 | 64.4 | 85.0 | 63.2 | 97.0 | 86.3 | 36.6 | 51.0 | 78.5 | 87.5 | 68.5 | 74.0 | 34.3 | 30.6 | 33.2 | 55.4 | 12.5 | 20.0 | 9.6 | 19.2 | 57.6 |
| BitFit | 0.10 | 72.8 | 87.0 | 59.2 | 97.5 | 85.3 | 59.9 | 51.4 | 78.7 | 91.6 | 72.9 | 69.8 | 61.5 | 55.6 | 32.4 | 55.9 | 66.6 | 40.0 | 15.7 | 25.1 | 65.2 |
| VPT-Shallow | 0.06 | 77.7 | 86.9 | 62.6 | 97.5 | 87.3 | 74.5 | 51.2 | 78.2 | 92.0 | 75.6 | 72.9 | 50.5 | 58.6 | 40.5 | 67.1 | 68.7 | 36.1 | 20.2 | 34.1 | 67.8 |
| VPT-Deep | 0.53 | 78.8 | 90.8 | 65.8 | 98.0 | 88.3 | 78.1 | 49.6 | 81.8 | 96.1 | 83.4 | 68.4 | 68.5 | 60.0 | 46.5 | 72.8 | 73.6 | 47.9 | 32.9 | 37.8 | 72.0 |
| Adapter | 0.16 | 69.2 | 90.1 | 68.0 | 98.8 | 89.9 | 82.8 | 54.3 | 84.0 | 94.9 | 81.9 | 75.5 | 80.9 | 65.3 | 48.6 | 78.3 | 74.8 | 48.5 | 29.9 | 41.6 | 73.9 |
| AdaptFormer | 0.16 | 70.8 | 91.2 | 70.5 | 99.1 | 90.9 | 86.6 | 54.8 | 83.0 | 95.8 | 84.4 | 76.3 | 81.9 | 64.3 | 49.3 | 80.3 | 76.3 | 45.7 | 31.7 | 41.1 | 74.7 |
| LoRA | 0.34 | 69.6 | 92.5 | 70.5 | 99 | 90.1 | 89.9 | 53.7 | 85.9 | 95.8 | 87.5 | 75.8 | 82.9 | 64.5 | 53.4 | 81.3 | 82.7 | 48.1 | 31.8 | 36.8 | 75.7 |
| MLAE | 0.34 | 65.7 | 90.7 | 70.4 | 99.1 | 91.1 | 86.4 | 53.9 | 82.4 | 94.9 | 86 | 74.9 | 77.5 | 64.5 | 50.2 | 79.9 | 75.1 | 43.3 | 27.7 | 35.8 | 73.6 |
| PISSA | 0.31 | 69.4 | 92.3 | 72.4 | 99.0 | 91.3 | 89.6 | 54.6 | 86.7 | 95.8 | 86.4 | 75.7 | 81.8 | 65.9 | 54.1 | 80.4 | 81.0 | 45.0 | 29.8 | 41.4 | 75.8 |
| AdaLoRA | 0.37 | 70.5 | 90.3 | 71.9 | 99.1 | 91.4 | 87 | 56 | 83.9 | 95.3 | 85.6 | 75.2 | 81.6 | 68.1 | 51.8 | 80.2 | 75.3 | 49.1 | 28.9 | 41.1 | 75.1 |
| DoRA | 0.34 | 70.0 | 91.9 | 71.3 | 99 | 90.0 | 88.7 | 55.5 | 85.4 | 95.5 | 86.8 | 76.5 | 82.5 | 66.5 | 53.0 | 80.3 | 82.5 | 46.8 | 30.1 | 41.9 | 75.8 |
| NOAH | 0.36 | 69.6 | 92.7 | 70.2 | 99.1 | 90.4 | 86.1 | 53.7 | 84.4 | 95.4 | 83.9 | 75.8 | 82.8 | 68.9 | 49.9 | 81.7 | 81.8 | 48.3 | 32.8 | 44.2 | 75.5 |
| FacT | 0.07 | 70.6 | 90.6 | 70.8 | 99.1 | 90.7 | 88.6 | 54.1 | 84.8 | 96.2 | 84.5 | 75.7 | 82.6 | 68.2 | 49.8 | 80.7 | 80.8 | 47.4 | 33.2 | 43.0 | 75.6 |
| SSF | 0.24 | 69.0 | 92.6 | 75.1 | 99.4 | 91.8 | 90.2 | 52.9 | 87.4 | 95.9 | 87.4 | 75.5 | 75.9 | 62.3 | 53.3 | 80.6 | 77.3 | 54.9 | 29.5 | 37.9 | 75.7 |
| FlexLoRA | 0.34 | 71.2 | 92.5 | 72.1 | 99.1 | 90.7 | 90.6 | 54.7 | 86.4 | 96.0 | 87.5 | 76 | 81.9 | 65.6 | 53.9 | 80.2 | 80.6 | 45.6 | 29.9 | 41.1 | 76.0 |

