# OpenReview forum: "FlexLoRA: Entropy-Guided Flexible Low-Rank Adaptation"
_ICLR.cc/2026/Conference — ICLR 2026 Poster_

### Official Review · Reviewer_eq8b · 2025-10-18

**Soundness:** 2
**Presentation:** 3
**Contribution:** 2
**Rating:** 2
**Confidence:** 4

**Summary:**

This paper proposes FlexLoRA, an entropy-guided dynamic low-rank adaptation method for parameter-efficient fine-tuning of large models. Unlike standard LoRA with fixed rank, FlexLoRA supports both rank pruning and expansion under a global parameter budget. It evaluates matrix importance using spectral entropy, enabling flexible capacity reallocation across layers. Additionally, a zero-impact initialization strategy ensures stable training when adding new rank directions. Experiments show that FlexLoRA consistently outperforms existing dynamic LoRA variants under the same parameter budget.

**Strengths:**

1. The paper addresses the key limitation of LoRA’s fixed rank by introducing a novel bidirectional rank adjustment mechanism that supports both pruning and expansion under a global budget. It also proposes a new matrix-level entropy-guided importance metric, offering a new perspective on evaluating low-rank matrices importance.
2. The paper presents comprehensive experiments across multiple domains, including Natural Language Understanding, Visual Recognition, and Commonsense Reasoning, along with extensive ablation studies.

**Weaknesses:**

1. Although the paper introduces a bidirectional rank adjustment mechanism supporting both pruning and expansion, the idea — if we ignore the proposed importance metric — conceptually resembles a combination of AdaLoRA (for adaptive pruning) and IncreLoRA [1] (for rank expansion). Hence, its novelty may be somewhat limited at the framework level.
2. Despite the conceptual soundness, the experimental credibility of the paper is questionable.
   - The tables do not clearly indicate which results are reproduced and which are cited from previous works, making it difficult to assess fairness. Moreover, no detailed experimental settings are provided in either the main text or the appendix, hindering reproducibility and leaving uncertainty about how baseline methods were implemented.
   - The improvements on commonsense reasoning tasks are marginal (e.g., LoRA 85.4% vs. FlexLoRA 85.5%), and given the high variance typically observed in such benchmarks, this raises doubts about the claimed effectiveness.
   - For visual recognition, the reported performance appears lower than that of prior works such as MLAE [2] and GLoRA [3], yet the paper provides no explanation or comparison of configurations, leaving potential inconsistencies unresolved.
   - The ablation study is also incomplete — only a subset of GLUE benchmarks (CoLA, MRPC, RTE, STS-B) is reported instead of the full suite, which may indicate selective reporting of favorable results.

------

[1] IncreLoRA: Incremental Parameter Allocation Method for Parameter-Efficient Fine-tuning. https://arxiv.org/abs/2308.12043

[2] MLAE: Masked LoRA Experts for Visual Parameter-Efficient Fine-Tuning. https://arxiv.org/abs/2405.18897

[3] One-for-All: Generalized LoRA for Parameter-Efficient Fine-tuning. https://arxiv.org/abs/2306.07967

**Questions:**

Additional Questions and Suggestions:

1. Prior study [1] have shown that randomly masking certain parameters within the LoRA matrices can effectively reduce redundancy and improve performance. This raises an important question: how does such a temporary stochastic masking approach compare to the permanent structural adjustments (i.e., pruning or expansion) proposed in this paper? An experimental comparison between random masking and FlexLoRA’s bidirectional rank adjustment would help clarify the relative advantages and trade-offs in terms of performance, stability, and generalization.
2. The paper does not discuss the computational efficiency of FlexLoRA. Since the proposed method introduces additional operations for rank evaluation, pruning, and expansion, it likely incurs extra computational overhead. Given that the performance gains of FlexLoRA appear modest, it is crucial to report and compare the training and inference time costs across different methods to fairly assess their overall efficiency.

------
If the authors can address the above concerns in response, I am willing to raise my score.

[1] MLAE: Masked LoRA Experts for Visual Parameter-Efficient Fine-Tuning. https://arxiv.org/abs/2405.18897

---

> ### Author Response · Authors · 2025-11-20
> **Response to Reviewer eq8b**
>
> Dear Reviewer eq8b:
>
> We would like to first extend our sincere gratitude for your time and effort in evaluating our manuscript. Your thorough evaluation and insightful comments are greatly appreciated. We will address your questions point by point and hope to resolve your concerns effectively.
>
> ## **W1**
>
> Conceptually, AdaLoRA and IncreLoRA realize rank pruning and rank expansion respectivel, and FlexLoRA combines both capabilities.
> However, it does not diminish the novelty of our method.
> First, FlexLoRA is the first to unify both rank pruning and expansion within a coherent mechanism.
> Second, the key challenge lies not the bidirectional reallocation but the metric that drives it: prior methods rely on heuristic importance scores, whereas our entropy guided metric provides a principled and stable basis for allocation.
> Finally, our reallocation operates at the matrix level rather than parameter level, resulting in a more consistent structural adjustment.

---

> ### Author Response · Authors · 2025-11-20
> **Response to Reviewer eq8b**
>
> ## **W2**
>
> We sincerely thank the reviewer for pointing out areas that can be improved.
> Below we respond to each point.
>
> ### On clarity of cited results and missing experimental details
>
> The experimental results in our tables follow a consistent sourcing rule.
> In Table 2, the ChatGPT and fully fine-tuning(Full FT) baseline is taken from [1].
> In Table 3, the fully fine-tuning(Full FT) result is taken from [2].
> All other results are reproduced by us under the same experimental settings.
>
> All metrics, seeds, and per-task hyper-parameters are added in the newly added Appendix D.
>
> [1] Liu et al., DoRA: Weight-Decomposed Low-Rank Adaptation.
>
> [2] Jie et al., Revisiting the Parameter Efficiency of Adapters from the Perspective of Precision Redundancy.
>
> ### On marginal improvements for commonsense reasoning tasks
>
> When the rank is large (e.g., r=32), each matrix already has enough representational capacity, so all methods tend to converge to similar performance, which explains the small gap between LoRA (85.4%) and FlexLoRA (85.5%).
> However, the effectiveness of FlexLoRA becomes clear under a constrained budget: when r=8, FlexLoRA significantly outperforms LoRA (82.4% → 85.2%).
>
> ### On visual tasks performance
>
> For a fair comparison, we evaluate MLAE under our own experimental setup; as shown in the table below, FlexLoRA clearly outperforms MLAE.
> In contrast, GLoRA is a hybrid fine-tuning framework that integrates multiple techniques and operates under a substantially larger design space, making a direct comparison to FlexLoRA inappropriate.
>
> **VTAB Benchmark Results (Accuracy %)**
>
> | Method    | Param. | Cifar100 | Caltech101 | DTD  | Flower102 | Pets | SVHN | Sun397 | Camelyon | EuroSAT | Resisc45 | Retinopathy | Clevr-Count | Clevr-Dist | DMLab | KITTI-Dist | dSpr-Loc | dSpr-Ori | sNORB-Azim | sNORB-Ele | Avg. |
> |-----------|--------|----------|------------|------|-----------|------|------|--------|----------|---------|----------|-------------|-------------|------------|-------|------------|----------|----------|------------|-----------|------|
> | Full FT   | 327M   | 68.9     | 87.7       | 64.3 | 97.2      | 86.9 | 84.7 | 38.8   | 79.7     | 95.7    | 84.2     | 73.9        | 56.3        | 58.6       | 41.7  | 65.5       | 57.5     | 46.7     | 25.7       | 29.1      | 68.9 |
> | MLAE      | 1.11M  | 49.0     | 87.1       | 62.4 | 97.9      | 88.1 | 69.8 | 47.6   | 79.9     | 94.4    | 80.0     | 70.1        | 73.5        | 57.4       | 42.3  | 74.0       | 69.2     | 43.2     | 13.2       | 30.2      | 66.6 |
> | FlexLoRA  | 1.18M  | 58.2     | 88.9       | 64.3 | 98.5      | 89.3 | 73.7 | 51.3   | 80.0     | 93.4    | 80.3     | 73.4        | 66.9        | 60.0       | 41.0  | 77.8       | 64.7     | 39.0     | 13.6       | 29.9      | 67.8 |
>
> ### On incomplete ablation
>
> We report ablations on a subset of GLUE tasks because they cover diverse linguistic phenomena.
> We agree that presenting the full GLUE suite is beneficial.
> As shown in the following table, the conclusions of our ablation study remain unchanged.
>
> | Method | Params | CoLA | MNLI | MRPC | RTE | QNLI | SST-2 | STS-B | QQP | Avg |
> |--------|--------|------|------|------|------|------|-------|-------|------|------|
> | Nuclear | 1.9M | 69.8 | 89.1 | 90.2 | 84.8 | 94.4 | 94.8 | 91.1 | 87.5 | 87.7 |
> | Frobenius | 1.9M | 69.4 | 88.8 | 86.5 | 86.3 | 94.1 | 94.7 | 90.8 | 86.4 | 87.1 |
> | Prune-only | 1.9M | 66.8 | 89.4 | 90.4 | 85.9 | 94.2 | 94.5 | 91.4 | 87.6 | 87.5 |
> | Expand-only | 1.9M | 68.3 | 89.5 | 89.5 | 87.7 | 94.3 | 93.8 | 91.3 | 86.5 | 87.6 |
> | Small-init | 1.9M | 69.2 | 89.4 | 89.7 | 85.9 | 94.4 | 95.2 | 91.2 | 87.5  | 87.8 |
> | Zero-init | 1.9M | 66.4 | 89.5 | 88.2 | 84.8 | 94.4 | 94.6 | 91.1 | 87.4 | 87.1 |
> | Orthogonal-init | 1.9M | 70.4 | 89.8 | 90.0 | 87.0 | 94.2 | 94.7 | 90.7 | 87.5 | 88.0 |
> | FlexLoRA | 1.9M | 71.8 | 90.0 | 90.9 | 88.8 | 94.2 | 95.2 | 91.5 | 90.3 | 89.1 |

---

> ### Author Response · Authors · 2025-11-20
> **Response to Reviewer eq8b**
>
> ## **Q1**
>
> Following the your suggestion, we directly compare FlexLoRA with random masking under the same experimental setting on VTAB benchmark.
> As the results shown below, FlexLoRA’s bidirectional rank adjustment outperforms random masking, demonstrating the benefit of our method.
>
> | Method    | Param. | Cifar100 | Caltech101 | DTD  | Flower102 | Pets | SVHN | Sun397 | Camelyon | EuroSAT | Resisc45 | Retinopathy | Clevr-Count | Clevr-Dist | DMLab | KITTI-Dist | dSpr-Loc | dSpr-Ori | sNORB-Azim | sNORB-Ele | Avg. |
> |-----------|--------|----------|------------|------|-----------|------|------|--------|----------|---------|----------|-------------|-------------|------------|-------|------------|----------|----------|------------|-----------|------|
> | Full FT   | 327M   | 68.9     | 87.7       | 64.3 | 97.2      | 86.9 | 84.7 | 38.8   | 79.7     | 95.7    | 84.2     | 73.9        | 56.3        | 58.6       | 41.7  | 65.5       | 57.5     | 46.7     | 25.7       | 29.1      | 68.9 |
> | MLAE      | 1.11M  | 49.0     | 87.1       | 62.4 | 97.9      | 88.1 | 69.8 | 47.6   | 79.9     | 94.4    | 80.0     | 70.1        | 73.5        | 57.4       | 42.3  | 74.0       | 69.2     | 43.2     | 13.2       | 30.2      | 66.6 |
> | FlexLoRA  | 1.18M  | 58.2     | 88.9       | 64.3 | 98.5      | 89.3 | 73.7 | 51.3   | 80.0     | 93.4    | 80.3     | 73.4        | 66.9        | 60.0       | 41.0  | 77.8       | 64.7     | 39.0     | 13.6       | 29.9      | 67.8 |
>
>
> ## **Q2**
>
> We have added full system-level metrics, including peak GPU memory, FLOPs, train runtime and eval runtime.
> As shown in the tables, FlexLoRA does not introduce noticeable GPU memory or FLOP overhead compared to LoRA and AdaLoRA.
> Our relevant system-level metrics outperformed AdaLoRA and achieved better results in most cases.
>
> **Comparison of training cost on NLU tasks**
>
> | Dataset      | Metric                 | FlexLoRA     | AdaLoRA     | LoRA        |
> |--------------|------------------------|--------------|-------------|-------------|
> |   CoLA       | Train Runtime          | 2162.6 s     | 2167.9 s    | 2117.9 s    |
> |              | GPU Mem                | 17446 MB     | 17487 MB    | 17443 MB    |
> |              | Total FLOPs            | $4.88×10^{16}$  | $4.90×10^{16}$ | $4.88×10^{16}$ |
> |              | Eval Runtime           | 9.3231 s     | 9.2842 s    | 10.4523 s   |
> |   RTE        | Train Runtime          | 1597.4 s     | 1574.9 s    | 1534.5 s    |
> |              | GPU Mem                | 17446 MB     | 17487 MB    | 17443 MB    |
> |              | Total FLOPs            | $3.55×10^{16}$  | $3.56×10^{16}$ | $3.55×10^{16}$ |
> |              | Eval Runtime           | 2.4398 s     | 2.3954 s    | 2.6691 s    |
>
> **Comparison of training cost on commmonsense reasoning tasks**
>
> | Model        | Metric                 | FlexLoRA     | AdaLoRA     | LoRA        |
> |--------------|------------------------|--------------|-------------|-------------|
> | LLaMA3-8B    | Train Runtime          | 6.01h        | 6.47h       | 4.76h       |
> |              | GPU Mem                | 38.0GB       | 43.3GB      | 37.6GB      |
>
> **Comparison of training cost on visual tasks**
>
> | Dataset      | Metric                 | FlexLoRA     | AdaLoRA     | LoRA        |
> |--------------|------------------------|--------------|-------------|-------------|
> | Cifar100 | Train Runtime          | 650 s        | 654 s       | 581 s       |
> |              | GPU Mem                | 8414 MB      | 8467 MB     | 7787 MB     |
> |              | Total FLOPs            | $5.62×10^{18}$  | $5.64×10^{18}$   | $5.63×10^{18}$   |
> |              | Eval Runtime           | 29.94 s      | 30.22 s     | 27.18 s     |
> | Resisc45 | Train Runtime          | 646 s        | 647 s       | 566 s       |
> |              | GPU Mem               |  8439 MB      | 8449 MB     | 7785 MB     |
> |              | Total FLOPs            | $5.62×10^{18}$  | $5.64×10^{18}$ | $5.63×10^{18}$ |
> |              | Eval Runtime           | 19.74 s      | 19.78 s     | 17.82 s     |
> | DMLab    | Train Runtime          | 695 s        | 699 s       | 774 s       |
> |              | GPU Mem                | 8451 MB      | 8465 MB     | 7785 MB     |
> |              | Total FLOPs            | $5.62×10^{18}$  | $5.64×10^{18}$ | $5.63×10^{18}$ |
> |              | Eval Runtime           | 67.06 s      | 66.98 s     | 59.91 s     |

---

> > ### Comment · Reviewer_eq8b · 2025-11-23
> >
> > Thank you for your response and for investing substantial effort in addressing several of my earlier concerns. However, several concerns still remain unresolved:
> >
> > 1. The performance improvement offered by FlexLoRA is limited. Although the authors highlight that “the effectiveness of FlexLoRA becomes clear under a constrained budget (e.g., when r = 8, FlexLoRA significantly outperforms LoRA: 82.4% → 85.2%),” in most practical scenarios practitioners can generally afford the memory overhead of using r = 32, which typically yields stronger fine-tuning performance. This makes it difficult to judge the practical benefits of FlexLoRA in commonly adopted settings.
> >
> > 2. In addition, the MLAE results reproduced by the authors are substantially lower than those reported in the original paper. I would appreciate it if the authors could provide a detailed description of their experimental configuration, including the expert drop rate, scaling coefficients, learning rate, weight decay, number of training epochs, and any other relevant hyperparameters used in their implementation.
> >
> > I look forward to the authors’ clarification on these points.

---

> ### Author Response · Authors · 2025-11-28
>
> Dear Reviewer eq8b:
>
> We sincerely thank the reviewer for the insightful comments. Below we address each point in detail.
>
> ## **On the practical benefit of FlexLoRA when r = 32**
>
> In practice, when the budget is sufficiently large (e.g., r = 32), PEFT methods are already very close to fully fine-tuning, making additional improvements inherently limited.
> Even under this saturated regime, FlexLoRA still achieves state-of-the-art results and remains only 0.6 points below fully fine-tuning, demonstrating that our entropy-guided allocation continues to provide benefits even when the performance ceiling is high.
>
> Moreover, as shown in the table below, AdaLoRA’s performance degrades at r = 32, whereas FlexLoRA further improves, highlighting that naïvely increasing rank does not guarantee better performance and that FlexLoRA’s dynamic allocation remains effective in high-budget settings.
>
> | Method  | Params.  |  BoolQ |  PIQA  |  SIQA  | HellaSwag | WinoGrande | ARC-e | ARC-c | OBQA | Avg.|
> |-------- |--------  |--------|--------|--------|--------|--------|--------|--------|--------|--------|
> |Fully Fine-tuning|    8B  |  75.3  |  89.9  |  81.5  |  95.8  |  87.6  |  91.6  |  79.3  |  87.4  |  86.1  |
> | $LoRA_{r=32}$    |  56.6M   |  75.6  |  89.5  |  81.2  |  95.1  |  85.1  |  80.1  |  90.3  |  86.2  |  85.4  |
> | $AdaLoRA_{r=32}$ |  56.6M   |  71.3  |  88.7  |  80.1  |  94.5  |  86.2  |  78.8  |  90.2  |  85.8  |  84.5  |
> | $FlexLoRA_{r=32}$|  56.6M   |  72.8  |  89.1  |  80.7  |  96.0  |  86.4  |  81.3  |  90.8  |  87.2  |  85.5  |

---

> > ### Author Response · Authors · 2025-11-28
> >
> > ## **On the MLAE reproduction gap**
> >
> > Our initial experiments were conducted under a setup that differs from MLAE not only in the underlying framework but also in data preprocessing. Moreover, we inserted LoRA blocks into six modules (QKV, attention.output.dense, mlp.fc1, mlp.fc2).
> > These together lead to lower accuracy for all PEFT methods.
> >
> > To address your concern, we have now switched to a timm-based implementation and use ViT-B/16 pre-trained on supervised ImageNet-21K as the default backbone. LoRA and FlexLoRA are applied to Q and V, and MLAE is applied to the QKV layers following the MLAE paper. Under this setting, all methods gain improvement as the table shown, and MLAE reaches 75.7, whereas FlexLoRA achieves 75.8, still maintaining an advantage. The complete implementation is included in the supplementary material.
> >
> > | Method    | Param. | Cifar100 | Caltech101 | DTD  | Flower102 | Pets | SVHN | Sun397 | Camelyon | EuroSAT | Resisc45 | Retinopathy | Clevr-Count | Clevr-Dist | DMLab | KITTI-Dist | dSpr-Loc | dSpr-Ori | sNORB-Azim | sNORB-Ele | Avg. |
> > |-----------|--------|----------|------------|------|-----------|------|------|--------|----------|---------|----------|-------------|-------------|------------|-------|------------|----------|----------|------------|-----------|------|
> > | $LoRA_{r=8}$   | 0.34M   |   69.3      |      91.2      |      70.6      |      99.1      |      91.2      |      90.3      |      54      |      85.1      |      95.9      |      87.5      |      74.9      |      82.6      |      63.7      |      52.3      |      81      |      79.3      |      48.3      |      32.2      |      42.8      |  75.6 |
> > | $MLAE_{r=8}$      | 0.44M  |    71.2      |      92.5      |      72.3      |      99.2      |      91.4      |      90.7      |      54.8      |      85      |      95.9      |      87.4      |      73.6      |      82.1      |      63.7      |      52.1      |      81.4      |      77.8      |      48.6      |      31.6      |      41.6      | 75.7 |
> > | $FlexLoRA_{r=8}$  | 0.34M  |   71.3      |      91.7      |      72.3      |      99      |      90.9      |      90.4      |      54.4      |      85.1      |      96.1      |      87.7      |      76.3      |      81.5      |      65.3      |      51.4      |      80.3      |      79      |      48      |      31.6      |      40.7 | 75.8 |
> >
> >
> > The learning rate and weight decay used in this experiment are shown in the table below. We use α = 2, dpr = 0.1 and epoch=100 for all methods, and MLAE’s dropout probability follows the values reported in Appendix A.2 of the original MLAE paper.
> >
> > |  Hyper-parameter    |  Cifar100 | Caltech101 | DTD  | Flower102 | Pets | SVHN | Sun397 | Camelyon | EuroSAT | Resisc45 | Retinopathy | Clevr-Count | Clevr-Dist | DMLab | KITTI-Dist | dSpr-Loc | dSpr-Ori | sNORB-Azim | sNORB-Ele |
> > |-----------|--------|----------|------------|------|-----------|------|------|--------|----------|---------|----------|-------------|-------------|------------|-------|------------|----------|----------|------------|
> > |learning rate|3e-3|6e-4|6e-4|2e-3|1e-3|2e-3|2e-3|6e-4|2e-3|2e-3|3e-3|2e-3|2e-3|7e-4|2e-3|6e-4|2e-3|2e-3|6e-4|
> > |weight decay|3e-3|1e-4|1e-4|5e-4|5e-3|5e-4|5e-4|1e-4|5e-4|5e-4|3e-4|5e-4|5e-4|5e-4|5e-4|1e-4|5e-4|5e-4|1e-4|
> >
> >
> > We agree that MLAE is a strong and well-designed baseline, and we have incorporated it into the main text of the revised paper.

---

> > > ### Comment · Reviewer_eq8b · 2025-11-28
> > >
> > > Thank you for the authors’ response. Regarding the practical benefit of FlexLoRA, I believe that if the method does not demonstrate a clear advantage over the widely adopted LoRA baseline under a practically acceptable rank setting, it will be difficult for this approach to be used in real applications. The community has accumulated substantial experience in tuning the hyperparameters of LoRA, and practitioners generally prefer methods that are already well understood and reliable. Therefore, without a significant improvement under realistic configurations, it is unlikely that FlexLoRA will be broadly adopted.
> > >
> > > With respect to the reproduction of MLAE, I acknowledge that the authors’ current experimental setup is relatively fair. However, I would like to ask how the initialization coefficients in the MLAE method were chosen in the reproduction. In addition, the authors appear to use dataset-specific learning rates and weight decay values, whereas the original MLAE paper seems to apply the same learning rate and weight decay across all datasets. I would appreciate it if the authors could clarify the basis for these choices. Furthermore, I suggest restricting MLAE to be applied only to the query and key matrices in order to ensure a more equitable comparison.
> > >
> > > For the reasons stated above, I am currently willing to raise my score to 4. However, I remain concerned about the practical usability of FlexLoRA. If the authors can address these concerns, I would be willing to further increase my score to 6 or 8.

---

> > > > ### Author Response · Authors · 2025-12-02
> > > >
> > > > Dear Reviewer eq8b:
> > > >
> > > > We sincerely thank you for your timely and thoughtful follow-up. We are glad that our previous response was able to resolve part of your concerns. Below we address the remaining points in detail.
> > > >
> > > > ## **On the practical benefit of FlexLoRA**
> > > >
> > > > We understand your concern: if a method does not show clear gains over a strong baseline like LoRA at r = 32, practitioners may hesitate to switch. However, solely treating r = 32 as “practically acceptable” setting is not reflective of real deployments. When GPU memory must be shared across multiple adapters or tasks, practitioners typically operate in the r = 4–16 range. In these resource-constrained settings, FlexLoRA offers clear practical value, as its r = 8 variant already matches the performance of LoRA at r = 32 (85.2% vs. 85.4%) while using less than half the parameters.
> > > >
> > > > Moreover, FlexLoRA’s benefit is not limited to commonsense reasoning. On NLU (GLUE) and visual tasks (VTAB), it consistently outperforms LoRA under the same rank budget, demonstrating practical gains across modalities.
> > > >
> > > > ## **On the reproduction of MLAE**
> > > >
> > > > Regarding the reproduction of MLAE, we clarify that in our initial experiments we treated the initialization coefficients in MLAE as a trainable variant of the LoRA scaling factor (alpha/r) and set them uniformly to 2 to ensure consistency across methods.
> > > > The dataset-specific learning rates and weight decay values were used because different VTAB datasets vary significantly in scale and difficulty, and therefore require different levels of regularization to avoid underfitting or overfitting.
> > > >
> > > > Nevertheless, to eliminate any possible confounding factor, we conducted an additional experiment where all methods (LoRA, MLAE, and FlexLoRA) use a fully unified configuration (learning rate = 2e-3, weight decay = 1e-3) across all datasets.
> > > > All initialization coefficients and MLAE dropout probabilities strictly follow the values in Appendix A.2 of the MLAE paper.
> > > > Furthermore, following the reviewer’s suggestion, we restricted MLAE to the query–key matrices to ensure a better comparison.
> > > >
> > > > As shown in the table below, FlexLoRA achieves the highest overall score (76.0), outperforming both MLAE and LoRA under this standardized setting.
> > > > These results confirm that the improvements of FlexLoRA do not arise from hyper-parameter choices but from the method itself.
> > > >
> > > > | Method    | Param. | Cifar100 | Caltech101 | DTD  | Flower102 | Pets | SVHN | Sun397 | Camelyon | EuroSAT | Resisc45 | Retinopathy | Clevr-Count | Clevr-Dist | DMLab | KITTI-Dist | dSpr-Loc | dSpr-Ori | sNORB-Azim | sNORB-Ele | Avg. |
> > > > |-----------|--------|----------|------------|------|-----------|------|------|--------|----------|---------|----------|-------------|-------------|------------|-------|------------|----------|----------|------------|-----------|------|
> > > > |LoRA|0.34M|69.6|92.5|70.5|99|90.1|89.9|53.7|85.9|95.8|87.5|75.8|82.9|64.5|53.4|81.3|82.7|48.1|31.8|36.8|75.7|
> > > > |MLAE|0.34M|65.7|90.7|70.4|99.1|91.1|86.4|53.9|82.4|94.9|86|74.9|77.5|64.5|50.2|79.9|75.1|43.3|27.7|35.8|73.6|
> > > > |FlexLoRA|0.34M|71.2|92.5|72.1|99.1|90.7|90.6|54.7|86.4|96.0|87.5|76|81.9|65.6|53.9|80.2|80.6|45.6|29.9|41.1|76.0|

---

### Official Review · Reviewer_SzUu · 2025-10-21

**Soundness:** 2
**Presentation:** 3
**Contribution:** 2
**Rating:** 6
**Confidence:** 4

**Summary:**

In this work, the authors propose to evolve the rank of low-rank adapters using entropic regularization on the singular value set. The framework allows both to reduce and to inflate the rank of the adapters, enabling dynamic reallocation across layers.
The proposed method is backed up by a variety of different numerical results.

**Strengths:**

The work is well written and clear. The proposed method, despite being very simple, seems to be effective against a variety of different baselines.

**Weaknesses:**

In the presentation of the algorithm, it is not clear which metric is precisely used to rank the importance of the single matrices. I would suggest that the authors include a more precise discussion about this in the revised version.

The proposed method is not able to significantly reduce the number of trainable parameters with respect to other methods like AdaLoRA or GeoLoRA [1]. It is also not clear how the global budget is allocated and kept under the maximal one, as the effect of inflating or not in terms of parameters depends on the layer one is considering.

[1] S. Schotthoefer et al., GeoLoRA: Geometric integration for parameter-efficient fine-tuning, ICLR 2025.

**Questions:**

1. From what I understood and from the numerical experiments, the proposed method should not be able to significantly reduce the number of trainable parameters. Is this correct?

For the rest, I would appreciate a comment from the authors about the "weaknesses" section.

---

> ### Author Response · Authors · 2025-11-20
> **Response to Reviewer SzUu**
>
> Dear Reviewer SzUu:
>
> We would like to first extend our sincere gratitude for your time and effort in evaluating our manuscript. Your thorough evaluation and insightful comments are greatly appreciated. We will address your questions point by point and hope to resolve your concerns effectively.
>
> ## **W1**
>
> The importance metric used in FlexLoRA is based on the spectral energy entropy of each matrix.
> During each reallocation step, we select the b matrices with the lowest entropy and remove one rank by pruning the smallest singular value, and select the b matrices with the highest entropy and add one rank using a zero-impact initialization.
>
> ## **W2 & Q1**
>
> Your understanding is correct. The goal of FlexLoRA is not to reduce the number of trainable parameters like GeoLoRA, but to better allocate ranks across matrices under a global rank budget. Since the total rank is kept constant throughout training, reallocating ranks among different matrices does not necessarily decrease the overall parameter count.
>
> As shown in the following table, although $FlexLoRA_{r=8}$ uses less than half the parameters of $LoRA_{r=32}$ , yet its performance differs by only 0.02% on average under the same experimental setting.
>
> **Partial Results on Commonsense Reasoning**
>
> | Method        | Param. | BoolQ | PIQA | SIQA | HellaS. | WinoG. | ARC-e | ARC-c | OBQA | Avg. |
> |---------------|--------|-------|------|------|---------|--------|-------|-------|------|------|
> | $LoRA_{r=32}$    | 56.6M  | 75.6  | 89.5 | 81.2 | 95.1    | 85.1   | 80.1  | 90.3  | 86.2 | 85.4 |
> | $FlexLoRA_{r=8}$ | 21.2M  | 74.3  | 88.6 | 81.0 | 95.6    | 84.9   | 80.2  | 90.7  | 86.4 | 85.2 |

---

> > ### Comment · Reviewer_SzUu · 2025-11-24
> >
> > First of all, I thank the authors for their rebuttal.
> >
> > Personally, my questions have been answered. For this reason, I am inclined to keep my original score.
> > In the meantime, I will continue to follow the discussion with the other reviewers.

---

> > > ### Author Response · Authors · 2025-11-25
> > > **We are happy that your questions have been answered!**
> > >
> > > Dear Reviewer SzUu,
> > >
> > > Thanks for your kind response. **We are happy that your questions have been solved**. If you have any further comments, please let us know. Thanks.
> > >
> > > Regards from the authors.

---

### Official Review · Reviewer_Z3z8 · 2025-10-30

**Soundness:** 3
**Presentation:** 3
**Contribution:** 3
**Rating:** 6
**Confidence:** 3

**Summary:**

The paper introduces FlexLoRA, a method for dynamically adjusting LoRA rank across layers using a matrix-level spectral entropy score. It periodically prunes low-entropy matrices and expands high-entropy ones while keeping a global parameter budget. New directions are added with zero-impact initialization, so the model output is not disturbed at the moment of expansion. Experiments on NLP and vision benchmarks show consistent, if sometimes modest, gains over standard LoRA and prior adaptive methods. Ablations suggest that the entropy score, bidirectional allocation, and the initialization choice each contribute to the improvements.

**Strengths:**

1. A simple, matrix-level entropy score guides where to prune and where to add rank, avoiding noisy element-wise heuristics.
2. True bidirectional allocation with zero-impact initialization lets the model expand capacity safely while staying within a budget.
3. Strong, cross-domain results (NLP and vision) with clear ablations show each component—entropy, bidirectionality, initialization—adds value.

**Weaknesses:**

1. Entropy ignores magnitude (energy).
The spectral‑entropy score is scale‑invariant: a matrix with very small Λ but uniform spread can have high entropy and thus be prioritized for expansion, even if its absolute contribution is negligible. The paper briefly compares against Frobenius/nuclear norms (Table 4), but does not explore combined criteria (e.g., entropy × energy) or energy‑gated expansion. This is a conceptual gap given the stated goal of measuring “importance.”
2. Baseline anomalies & reporting details.
In Table 1, LoRA’s MRPC (68.4 Acc) and QQP (63.1 Acc) are unusually low for modern setups, reducing the average (81.7) and potentially inflating relative gains. GLUE typically reports MRPC/QQP F1 scores alongside, or instead of, accuracy. Please clarify the metrics, seeds, and hyperparameters per task, and include mean ± std across multiple seeds.
3. Cost/overhead not quantified.
Maintaining orthogonality (Eq. 2) and dynamically changing parameter shapes may incur compute/memory overhead vs. standard LoRA. No wall‑clock or throughput numbers are provided, especially on large LLM runs.
4. Limited theoretical justification linking entropy to “importance.”
Appendix B argues that the least singular value contributes least to entropy under certain conditions, which supports the pruning rule, but there’s no theoretical link to downstream loss reduction or generalization.

**Questions:**

See weakness

---

> ### Author Response · Authors · 2025-11-20
> **Response to Reviewer Z3z8**
>
> Dear Reviewer Z3z8:
>
> We would like to first extend our sincere gratitude for your time and effort in evaluating our manuscript. Your thorough evaluation and insightful comments are greatly appreciated. We will address your questions point by point and hope to resolve your concerns effectively.
>
> ## **W1**
>
> As shown in the newly added Appendix F, we directly inspected the singular value distribution of ten matrices after training on CoLA, and we did not observe the situation you are concerned about.
> Matrices with with very small Λ rarely exhibit high entropy, and thus are not mistakenly expanded in practice.
> However, we may not fully capture the exact meaning of the “combined criteria” you refer to.
> Based on our own understanding, we have implemented two energy-considering variants:
>
> $$
> I_{Elem} = -\frac{1}{r\log r} \sum_i \lambda_i s_i \log(s_i + \epsilon),
> s_i = \frac{\lambda_i^2}{\sum_j \lambda_j^2}
> $$
>
> $$
> I_{Mat} = -\frac{1}{r\log r} \left(\sum_i \lambda_i \right)\left(\sum_i s_i \log(s_i+\epsilon)\right),
> s_i = \frac{\lambda_i^2}{\sum_j \lambda_j^2}
> $$
>
> As shown in the following table, neither variant outperforms FlexLoRA, confirming that our spectral energy entropy is a more reliable metric for measuring the importance of the matrices.
>
> **Comparison of FlexLoRA and two variants on NLU tasks**
>
> | Method | Params | CoLA | MNLI | MRPC | RTE | QNLI | SST-2 | STS-B | QQP | Avg |
> |--------|--------|------|------|------|------|------|-------|-------|------|------|
> | Elem | 1.9M | 68.1 | 89.1 | 91.0 | 86.3 | 94.5 | 94.7 | 90.9 | 87.4 | 87.8 |
> | Mat | 1.9M | 69.0 | 89.5 | 89.3 | 83.8 | 94.6 | 95.5 | 91.8 | 87.2 | 87.6 |
> | FlexLoRA | 1.9M | 71.8 | 90.0 | 90.9 | 88.8 | 94.2 | 95.2 | 91.5 | 90.3 | 89.1 |
>
> ## **W2**
>
> Thank you for carefully examining the reported baselines and for pointing out the anomalously low LoRA results for MRPC and QQP in Table 1. We sincerely apologize for this recording error. In the updated version of the paper, we corrected the affected LoRA scores and ensure that all baselines are reported consistently.
>
> We have added all metrics, seeds, and per-task hyper-parameters in the newly added Appendix D. Our reported results are already the mean of 5 runs with different random seeds, and we additionally included the corresponding standard deviations for each method in the updated version of the paper.

---

> ### Author Response · Authors · 2025-11-20
> **Response to Reviewer Z3z8**
>
> ## **W3**
>
> We have added full system-level metrics, including peak GPU memory, FLOPs, train runtime and eval runtime.
> As shown in the tables, FlexLoRA does not introduce noticeable GPU memory or FLOP overhead compared to LoRA and AdaLoRA.
> Our relevant system-level metrics outperformed AdaLoRA and achieved better results in most cases.
>
> **Comparison of training cost on NLU tasks**
>
> | Dataset      | Metric                 | FlexLoRA     | AdaLoRA     | LoRA        |
> |--------------|------------------------|--------------|-------------|-------------|
> |   CoLA       | Train Runtime          | 2162.6 s     | 2167.9 s    | 2117.9 s    |
> |              | GPU Mem                | 17446 MB     | 17487 MB    | 17443 MB    |
> |              | Total FLOPs            | $4.88×10^{16}$  | $4.90×10^{16}$ | $4.88×10^{16}$ |
> |              | Eval Runtime           | 9.3231 s     | 9.2842 s    | 10.4523 s   |
> |   RTE        | Train Runtime          | 1597.4 s     | 1574.9 s    | 1534.5 s    |
> |              | GPU Mem                | 17446 MB     | 17487 MB    | 17443 MB    |
> |              | Total FLOPs            | $3.55×10^{16}$  | $3.56×10^{16}$ | $3.55×10^{16}$ |
> |              | Eval Runtime           | 2.4398 s     | 2.3954 s    | 2.6691 s    |
>
> **Comparison of training cost on commmonsense reasoning tasks**
>
> | Model        | Metric                 | FlexLoRA     | AdaLoRA     | LoRA        |
> |--------------|------------------------|--------------|-------------|-------------|
> | LLaMA3-8B    | Train Runtime          | 6.01h        | 6.47h       | 4.76h       |
> |              | GPU Mem                | 38.0GB       | 43.3GB      | 37.6GB      |
>
> **Comparison of training cost on visual tasks**
>
> | Dataset      | Metric                 | FlexLoRA     | AdaLoRA     | LoRA        |
> |--------------|------------------------|--------------|-------------|-------------|
> | Cifar100 | Train Runtime          | 650 s        | 654 s       | 581 s       |
> |              | GPU Mem                | 8414 MB      | 8467 MB     | 7787 MB     |
> |              | Total FLOPs            | $5.62×10^{18}$  | $5.64×10^{18}$   | $5.63×10^{18}$   |
> |              | Eval Runtime           | 29.94 s      | 30.22 s     | 27.18 s     |
> | Resisc45 | Train Runtime          | 646 s        | 647 s       | 566 s       |
> |              | GPU Mem               |  8439 MB      | 8449 MB     | 7785 MB     |
> |              | Total FLOPs            | $5.62×10^{18}$  | $5.64×10^{18}$ | $5.63×10^{18}$ |
> |              | Eval Runtime           | 19.74 s      | 19.78 s     | 17.82 s     |
> | DMLab    | Train Runtime          | 695 s        | 699 s       | 774 s       |
> |              | GPU Mem                | 8451 MB      | 8465 MB     | 7785 MB     |
> |              | Total FLOPs            | $5.62×10^{18}$  | $5.64×10^{18}$ | $5.63×10^{18}$ |
> |              | Eval Runtime           | 67.06 s      | 66.98 s     | 59.91 s     |
>
> ## **W4**
>
> Our intuition follows the same foundation as effective rank: entropy characterizes how fully a matrix utilizes its available rank.
> By adopting normalized Frobenius-energy spectral entropy, we obtain a metric that reflects both the structure of the matrix and the distribution of usable directions.
> Low entropy means almost all energy is captured by a few top directions, so the matrix is less important and can be pruned.
> High entropy means many directions carry non-negligible energy, which indicates that the matrix has richer structural capacity and benefits more from additional rank, so the matrix is more important and should be expanded.
>
> The pruning rule in Appendix B only explains why the smallest singular value and relevant direction vectors are pruned and it is not intended to, nor should it, linked to downstream loss reduction or generalization.

---

### Official Review · Reviewer_uSv1 · 2025-11-01

**Soundness:** 2
**Presentation:** 2
**Contribution:** 2
**Rating:** 2
**Confidence:** 4

**Summary:**

This paper introduces an entropy-based metric to estimate the importance score of each weight-update matrix and proposes FlexLoRA, a method that dynamically adjusts the rank of LoRA modules by reducing the rank of those with lower importance scores while increasing that of the more important ones.

**Strengths:**

The framework is clear and intuitive. The authors conduct many experiments on GLUE, commonsense reasoning, and the Visual Task Adaptation Benchmarks, demonstrating the effectiveness of the proposed method.

**Weaknesses:**

1. The paper omits key system-level metrics such as peak GPU memory usage and FLOPs, which are essential for assessing efficiency.
2. The proposed entropy-guided importance metric is closely related to the effective rank concept from [1], limiting the novelty of the contribution.
3. The reported performance gains are marginal, particularly for large-scale models like LLaMA3-8B. Moreover, in Table 2, the number of trainable parameters in AdaLoRA is only about half that of the proposed model, making the comparison appear unfair.


[1] Olivier Roy and Martin Vetterli, The effective rank: A measure of effective dimensionality.

**Questions:**

Since the framework adjusts the rank dynamically during training, it would be valuable to report how this process affects the overall training time and computational cost.

---

> ### Author Response · Authors · 2025-11-20
> **Response to Reviewer uSv1**
>
> Dear Reviewer uSv1:
>
> We would like to first extend our sincere gratitude for your time and effort in evaluating our manuscript. Your thorough evaluation and insightful comments are greatly appreciated. We will address your questions point by point and hope to resolve your concerns effectively.
>
> ## **W1 & Q1**
>
> We have added full system-level metrics, including peak GPU memory, FLOPs, train runtime and eval runtime.
> As shown in the tables, FlexLoRA does not introduce noticeable GPU memory or FLOP overhead compared to LoRA and AdaLoRA.
> Our relevant system-level metrics outperformed AdaLoRA and achieved better results in most cases.
>
> **Comparison of training cost on NLU tasks**
>
> | Dataset      | Metric                 | FlexLoRA     | AdaLoRA     | LoRA        |
> |--------------|------------------------|--------------|-------------|-------------|
> |   CoLA       | Train Runtime          | 2162.6 s     | 2167.9 s    | 2117.9 s    |
> |              | GPU Mem                | 17446 MB     | 17487 MB    | 17443 MB    |
> |              | Total FLOPs            | $4.88×10^{16}$  | $4.90×10^{16}$ | $4.88×10^{16}$ |
> |              | Eval Runtime           | 9.3231 s     | 9.2842 s    | 10.4523 s   |
> |   RTE        | Train Runtime          | 1597.4 s     | 1574.9 s    | 1534.5 s    |
> |              | GPU Mem                | 17446 MB     | 17487 MB    | 17443 MB    |
> |              | Total FLOPs            | $3.55×10^{16}$  | $3.56×10^{16}$ | $3.55×10^{16}$ |
> |              | Eval Runtime           | 2.4398 s     | 2.3954 s    | 2.6691 s    |
>
> **Comparison of training cost on commmonsense reasoning tasks**
>
> | Model        | Metric                 | FlexLoRA     | AdaLoRA     | LoRA        |
> |--------------|------------------------|--------------|-------------|-------------|
> | LLaMA3-8B    | Train Runtime          | 6.01h        | 6.47h       | 4.76h       |
> |              | GPU Mem                | 38.0GB       | 43.3GB      | 37.6GB      |
>
> **Comparison of training cost on visual tasks**
>
> | Dataset      | Metric                 | FlexLoRA     | AdaLoRA     | LoRA        |
> |--------------|------------------------|--------------|-------------|-------------|
> | Cifar100 | Train Runtime          | 650 s        | 654 s       | 581 s       |
> |              | GPU Mem                | 8414 MB      | 8467 MB     | 7787 MB     |
> |              | Total FLOPs            | $5.62×10^{18}$  | $5.64×10^{18}$   | $5.63×10^{18}$   |
> |              | Eval Runtime           | 29.94 s      | 30.22 s     | 27.18 s     |
> | Resisc45 | Train Runtime          | 646 s        | 647 s       | 566 s       |
> |              | GPU Mem                | 8439 MB      | 8449 MB     | 7785 MB     |
> |              | Total FLOPs            | $5.62×10^{18}$  | $5.64×10^{18}$ | $5.63×10^{18}$ |
> |              | Eval Runtime           | 19.74 s      | 19.78 s     | 17.82 s     |
> | DMLab    | Train Runtime          | 695 s        | 699 s       | 774 s       |
> |              | GPU Mem                | 8451 MB      | 8465 MB     | 7785 MB     |
> |              | Total FLOPs            | $5.62×10^{18}$  | $5.64×10^{18}$ | $5.63×10^{18}$ |
> |              | Eval Runtime           | 67.06 s      | 66.98 s     | 59.91 s     |
>
> ## **W2**
>
> Although our metric is conceptually related to effective rank, it does not limit the novelty of our contribution. We are the first to use a entropy–based metric as a matrix-level importance score in PEFT, and it provides a more principled alternative to the heuristic measure in AdaLoRA. Moreover, unlike effective rank normalizing singular values by their sum, we normalize by their squared sum, producing a clearer characterization of the spectral energy distribution. Finally, effective rank lies in [0,r], which is difficult to compare across matrices, whereas our metric is normalized to [0,1], enabling fair cross-layer ranking. As shown in the table below, our FlexLoRA outperforms effective rank under the same experimental settings.
>
> **Effective rank results on NLU tasks**
>
> | Method   | Params | CoLA | MNLI | MRPC | RTE | QNLI | SST-2 | STS-B | QQP | Avg |
> |--------  |--------|------|------|------|------|------|-------|-------|------|------|
> | ERank    | 1.9M   | 69.7 | 89.4 | 90.3 | 83.8 | 94.4 | 95.1 | 91.5 | 87.7 | 87.7 |
> | FlexLoRA | 1.9M   | 71.8 | 90.0 | 90.9 | 88.8 | 94.2 | 95.2 | 91.5 | 90.3 | 89.1 |
>
> ## **W3**
>
> The comparison in Table 2 may have caused a misunderstanding: the fair comparison is between $AdaLoRA_{r=8}$ (21.2M params) and $FlexLoRA_{r=8}$ (21.2M params). Under similar parameter budgets, the average scores are 85.2 and 85.4, showing that FlexLoRA achieves better performance than AdaLoRA.

---

> > ### Comment · Reviewer_uSv1 · 2025-11-21
> >
> > Thank you for the detailed response.
> >
> > For W1: Please add more details regarding the setting used to measure system-level metrics, specifically the rank of each method.
> >
> > For W3: Referring to Table 2, the average scores for AdaLoRA and FlexLoRA at rank 8 are 85.1 and 85.2. Is there a typo in your text response, or is the data in the table incorrect?
> >
> > Additionally, I am concerned that while LoRA, AdaLoRA, and FlexLoRA are compared at rank 8, AdaLoRA is excluded from the rank 32 results. Why is this baseline missing?

---

> > > ### Author Response · Authors · 2025-11-23
> > >
> > > Dear Reviewer uSv1:
> > >
> > > We sincerely thank you for your response.
> > >
> > > ## **On Experimental Setting**
> > >
> > > All system-level metrics measurements are conducted under a consistent configuration, where all methods (LoRA, AdaLoRA, and FlexLoRA) are trained with rank = 8, batch size = 64, and identical training hyperparameters as described in Appendix D.
> > >
> > > ## **Clarification on Table 2**
> > >
> > > Thank you for catching this issue.
> > > The mismatch was a typo in our written response, not an error in the table.
> > > The correct results are exactly those shown in Table 2: AdaLoRA achieves 85.1 and FlexLoRA achieves 85.2 at rank 8.
> > > We sincerely apologize for our mistake.
> > >
> > > ## **On Missing ${AdaLoRA}_{r=32}$ Results**
> > >
> > > In preliminary experiments, AdaLoRA exhibited lower performance at r = 32 than at r = 8 on commonsense reasoning tasks, which led to its exclusion in the initial draft.
> > > To ensure completeness and fairness, we have included $AdaLoRA_{r=32}$ in the updated table.
> > > As the results in the table below shown, FlexLoRA continues to outperform both LoRA and AdaLoRA at r = 32.
> > >
> > >
> > > | Method  | Params.  |  BoolQ |  PIQA  |  SIQA  | HellaSwag | WinoGrande | ARC-e | ARC-c | OBQA | Avg.|
> > > |-------- |--------  |--------|--------|--------|--------|--------|--------|--------|--------|--------|
> > > | $LoRA_{r=32}$    |  56.6M   |  75.6  |  89.5  |  81.2  |  95.1  |  85.1  |  80.1  |  90.3  |  86.2  |  85.4  |
> > > | $AdaLoRA_{r=32}$ |  56.6M   |  71.3  |  88.7  |  80.1  |  94.5  |  86.2  |  78.8  |  90.2  |  85.8  |  84.5  |
> > > | $FlexLoRA_{r=32}$|  56.6M   |  72.8  |  89.1  |  80.7  |  96.0  |  86.4  |  81.3  |  90.8  |  87.2  |  85.5  |

---

### Author Response · Authors · 2025-11-20
**Common Response**

Dear AC and all reviewers,

We sincerely appreciate the reviewers’ time, effort, and insightful feedback on our work. The thoughtful comments have greatly helped us strengthen the clarity and quality of our manuscript. In response to these constructive suggestions, we have carefully revised the paper and addressed all concerns as detailed below:

1. In response to Reviewer uSv1, Weakness 1 and Question 1, Reviewer Z3z8, Weakness 3 and Reviewer eq8b, Question 2, we have added the comparison of training cost between FlexLoRA, AdaLoRA and LoRA in Appendix E.

2. In response to Reviewer Z3z8, Weakness 2 and Reviewer eq8b, Weakness 2, we have corrected the previously recorded incorrect data, added the standard deviation across multiple seeds, clarified which results were cited from previous works and added detailed experimental settings.
These updates are presented in Table 1 and Appendix D.

3. In response to Reviewer eq8b, Weakness 2, we have provided complete ablation study results on GLUE benchmarks in Table 4-6 and expanded the visual task experiment to include a comparison with MLAE, which is reflected in Table 3.

We hope that these revisions adequately address the reviewers’ concerns and help further clarify and strengthen our contributions.
We once again sincerely thank the reviewers for their constructive feedback and valuable guidance.

Sincerely,

Authors.

---

### Meta-Review · Area_Chair_7n5z · 2026-01-12

**Summary:**

This paper introduces aFlexLoRA, a method that dynamically adjusts the rank of LoRA modules based on entropy-based importance scores. Most concerns that were raised around more detailed system-level metrics, novelty and experimental results were addressed via comprehensive updates.

**Reviewer Concerns:**

uSv1> Raised 3 concerns: (1) paper omits key system-level metrics (2) limited novelty relative to [1] Roy and Vetterli and (3) marginal performance gains. These seem to have been addressed.

Z3z8> Raised 4 concerns:  (1) the magnitude-insensitivity of entropy, (2) anomalously low baselines and missing experimental details, (3) missing system-level metrics analysis, and (4) insufficient theoretical justification. These all also seem to have been addressed.

eq8b> raised six weaknesses: 1) limited novelty of the framework, (2) unclear reporting of results, (3) marginal improvements on CR tasks, (4) comparison with MLAE on visual tasks, (5) incomplete ablation study, and (6) missing system-level metrics analysis. Many of these concerns were raised

**Reviewer Scores:**

uSv1> would likely raise the score

eq8b> I am currently willing to raise my score to 4. However, I remain concerned about the practical usability of FlexLoRA. If the authors can address these concerns, I would be willing to further increase my score to 6 or 8.

Other two reviewers scored the paper at 6 and there is no reason to believe they would lower the score.

---

### Decision · Program_Chairs · 2026-01-26

Accept (Poster)